# OUT-OF-DISTRIBUTION GENERALIZATION VIA RISK EXTRAPOLATION

## ABSTRACT

Distributional shift is one of the major obstacles when transferring machine learning prediction systems from the lab to the real world. To tackle this problem, we assume that variation across training domains is representative of the variation we might encounter at test time, but also that *shifts at test time may be more extreme in magnitude*. In particular, we show that reducing differences in risk across training domains can reduce a model's sensitivity to a wide range of extreme distributional shifts, including the challenging setting where the input contains both causal and anti-causal elements. We motivate this approach, **Risk Extrapolation (REx)**, as a form of robust optimization over a perturbation set of extrapolated domains (MM-REx), and propose a penalty on the variance of training risks (V-REx) as a simpler variant. We prove that variants of REx can recover the causal mechanisms of the targets, while also providing some robustness to changes in the input distribution ("covariate shift"). By appropriately trading-off robustness to causally induced distributional shifts and covariate shift, REx is able to outperform alternative methods such as Invariant Risk Minimization in situations where these types of shift co-occur.

## 1 INTRODUCTION

While neural networks often exhibit super-human generalization on the training distribution, they can be extremely sensitive to distributional shift, presenting a major roadblock for their practical application (Su et al., 2019; Engstrom et al., 2017; Recht et al., 2019; Hendrycks & Dietterich, 2019). This sensitivity is often caused by relying on "spurious" features unrelated to the core concept we are trying to learn (Geirhos et al., 2018). For instance, Beery et al. (2018) give the example of an image recognition model failing to correctly classify cows on the beach, since it has learned to make predictions based on the features of the background (e.g. a grassy field) instead of just the animal.

In this work, we consider **out-of-distribution (OOD) generalization**, also known as **domain generalization**, where a model must generalize appropriately to a new test domain for which it has neither labeled nor unlabeled training data. Following common practice (Ben-Tal et al., 2009), we formulate this as optimizing the worst-case performance over a **perturbation set** of possible test domains, $\mathcal{F}$:

$$\mathcal{R}_{\mathcal{F}}^{\mathrm{OOD}}(\theta) = \max_{e \in \mathcal{F}} \mathcal{R}_e(\theta) \tag{1}$$

Since generalizing to arbitrary test domains is impossible, the choice of perturbation set encodes our assumptions about which test domains might be encountered. Instead of making such assumptions *a priori*, we assume access to data from multiple training domains, which can inform our choice of perturbation set. A classic approach for this setting is **group distributionally robust optimization (DRO)** (Sagawa et al., 2019), where $\mathcal{F}$ contains all mixtures of the training distributions. This is mathematically equivalent to considering convex combinations of the training *risks*.

However, we aim for a more ambitious form of OOD generalization, over a larger perturbation set. Our method **minimax Risk Extrapolation (MM-REx)** is an extension of DRO where $\mathcal{F}$ instead contains *affine* combinations of training risks, see Figure 1. Under specific circumstances, MM-REx can be thought of as DRO over a set of extrapolated domains[1], allowing us to carry over machinery

---

[1]We define "extrapolation" to mean "outside the convex hull", see Appendix B for more.

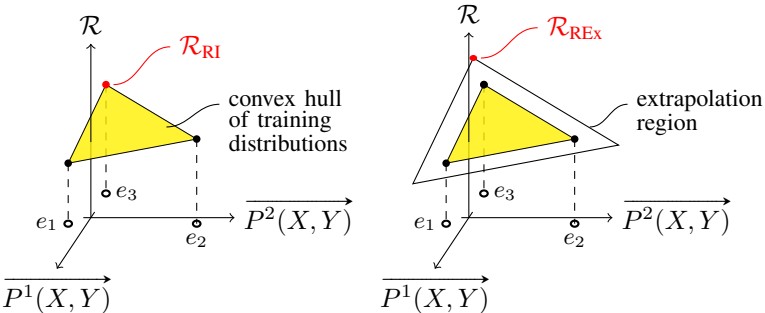

Figure 1: **Left**: Robust optimization optimizes worst-case performance over the convex hull of training distributions. **Right**: By extrapolating risks, REx encourages robustness to larger shifts. Here $e_1, e_2$, and $e_3$ represent training distributions, and $\overrightarrow{P^1(X,Y)}, \overrightarrow{P^2(X,Y)}$ represent some particular directions of variation in the affine space of quasiprobability distributions over $(X,Y)$.

developed for DRO, such as the $\mathcal{O}(1/\sqrt{T})$ convergence rate proven by Sagawa et al. (2019). But MM-REx also unlocks fundamental new generalization capabilities unavailable to DRO.

In particular, focusing on supervised learning, we show that Risk Extrapolation can uncover invariant relationships between inputs $X$ and targets $Y$. Intuitively, an **invariant relationship** is a statistical relationship which is maintained across all domains in $\mathcal{F}$. Returning to the cow-on-the-beach example, the relationship between the animal and the label is expected to be invariant, while the relationship between the background and the label is not. A model which bases its predictions on such an invariant relationship is said to perform **invariant prediction**.[2]

Many domain generalization methods assume $P(Y|X)$ is an invariant relationship, limiting distributional shift to changes in $P(X)$, which are known as **covariate shift** (David et al., 2010). This assumption can easily be violated, however. For instance, when $Y$ causes $X$, a more sensible assumption is that $P(X|Y)$ is fixed, with $P(Y)$ varying across domains (Schölkopf et al., 2012; Lipton et al., 2018). In general, invariant prediction may involve an aspect of causal discovery. Depending on the perturbation set, however, other, more predictive, invariant relationships may also exist (Koyama & Yamaguchi, 2020).

The first method for invariant prediction to be compatible with modern deep learning problems and techniques is **Invariant Risk Minimization (IRM)** (Arjovsky et al., 2019), making it a natural point of comparison. Our work focuses on explaining how REx addresses OOD generalization, and highlighting differences (especially advantages) compared with IRM and other domain generalization methods, see Table 1. Broadly speaking, REx optimizes for robustness to the forms of distributional shift that have been observed to have the largest impact on performance in training domains. This can prove a significant advantage over the more focused (but also limited) robustness that IRM targets. For instance, unlike IRM, REx can also encourage robustness to covariate shift (see Section 3).

And indeed, our experiments show that REx significantly outperforms IRM in settings that involve covariate shift and require invariant prediction, including modified versions of CMNIST and simulated robotics tasks from the Deepmind control suite. On the other hand, because REx does not distinguish between underfitting and inherent noise, IRM has an advantage in settings where some domains are intrinsically harder than others. We perform several other sets of experiments in order to better understand and compare REx and IRM. Our contributions include:

1) MM-REx, a novel domain generalization problem formulation suitable for invariant prediction.

2) Demonstrating that REx solves invariant prediction tasks where IRM fails due to covariate shift.

3) Proving that equality of risks can be a sufficient criteria for discovering causal structure.

---

[2]Note this is different from learning an invariant representation (Ganin et al., 2016); see Section 2.2.

| Method | Invariant Prediction | Cov. Shift Robustness | Suitable for Deep Learning |
|:---:|:---:|:---:|:---:|
| DRO | ✗ | ✓ | ✓ |
| (C-)ADA | ✗ | ✓ | ✓ |
| ICP | ✓ | ✗ | ✗ |
| IRM | ✓ | ✗ | ✓ |
| REx | ✓ | ✓ | ✓ |

Table 1: A comparison of approaches for OOD generalization.

## 2 BACKGROUND & RELATED WORK

We consider multi-source domain generalization, where our goal is to find parameters $\theta$ that perform well on unseen domains, given a set of $m$ training **domains**, $\mathcal{E} = \{e_1, .., e_m\}$, sometimes also called **environments**. We assume the loss function, $\ell$ is fixed, and domains only differ in terms of their data distribution $P_e(X, Y)$ and dataset $D_e$. The **risk function** for a given domain/distribution $e$ is:

$$\mathcal{R}_e(\theta) \doteq \mathbb{E}_{(x,y) \sim P_e(X,Y)} \ell(f_\theta(x), y) \tag{2}$$

We refer to members of the set $\{\mathcal{R}_e | e \in \mathcal{E}\}$ as the **training risks** or simply **risks**. Changes in $P_e(X, Y)$ can be categorized as either changes in $P(X)$ (**covariate shift**), changes in $P(Y|X)$ (**concept shift**), or a combination. The standard approach to learning problems is **Empirical Risk Minimization (ERM)**, which minimizes the average loss across all the training examples from all the domains:

$$\mathcal{R}_{\text{ERM}}(\theta) \doteq \mathbb{E}_{(x,y) \sim \cup_{e \in \mathcal{E}} D_e} \ell(f_\theta(x), y) = \sum_e \frac{1}{|D_e|} \mathbb{E}_{(x,y) \sim D_e} \ell(f_\theta(x), y) \tag{3}$$

### 2.1 ROBUST OPTIMIZATION

An approach more taylored to OOD generalization is **robust optimization** (Ben-Tal et al., 2009), which aims to optimize a model's worst-case performance over some **perturbation set** of possible data distributions, $\mathcal{F}$ (see Eqn. 1). When only a single training domain is available, it is common to assume that $P(Y|X)$ is fixed[3], and let $\mathcal{F}$ be all distributions within some $f$-divergence ball of the training $P(X)$ (Hu et al., 2016; Bagnell, 2005). As another example, adversarial robustness can be seen as instead using a Wasserstein ball as a perturbation set (Sinha et al., 2017).

In **multi-source domain generalization**, test distributions are often assumed to be mixtures (i.e. convex combinations) of the training distributions; this is equivalent to setting $\mathcal{F} \doteq \mathcal{E}$:

$$\mathcal{R}_{\text{RI}}(\theta) \doteq \max_{\substack{\Sigma_e \lambda_e = 1 \\ \lambda_e \geq 0}} \sum_{e=1}^m \lambda_e \mathcal{R}_e(\theta) = \max_{e \in \mathcal{E}} \mathcal{R}_e(\theta) . \tag{4}$$

We call this objective **Risk Interpolation (RI)**, or **(group) Domain Robust Optimization (DRO)**, following Sagawa et al. (2019). While single-source methods classically assume that the probability of each data-point can vary independently (Hu et al., 2016), DRO yields a much lower dimensional perturbation set, with at most one direction of variation per domain, regardless of the dimensionality of $X$ and $Y$. It can also provide robustness to any form of shift in $P(X, Y)$ which occurs across training domains, whereas single-source methods typically assume only $P(X)$ can change. Minimax REx is an extension of this approach to affine combinations of training risks.

### 2.2 INVARIANCE AND CAUSALITY

An **invariant predictor**, $\Phi$, is a function of $X$ with the property that $P_e(Y|\Phi)$ is equal $\forall e \in \mathcal{E}$. In other words, the relationship between such a $\Phi$ and $Y$ is invariant to the choice of domain. **Invariant relationships** between $X$ and $Y$ are those that can be written as $\hat{P}_\Phi(Y|X = x) \doteq P(Y|\Phi(x))$ with

---

[3]This *assumption* is often called "covariate shift", but we assume covariate/concept shift can co-occur.

---

**Algorithm 1** Variance Risk Extrapolation (V-REx)

---

**Require:** $D_1, ..., D_m$: training sets from different domains
**Require:** $\alpha, \beta$: learning rate and penalty term hyperparameters
 1: randomly initialize model parameters $\theta$
 2: **while** not done **do**
 3:    **for all** $D_i$ **do**
 4:       Estimate risk $\hat{\mathcal{R}}_i(\theta)$ using a minibatch of $K$ examples from $D_i$
 5:    **end for**
 6:    Update $\theta \leftarrow \theta - \alpha \nabla_\theta \left( \sum_i \hat{\mathcal{R}}_i(\theta) + \beta Var_i(\hat{\mathcal{R}}_i(\theta)) \right)$
 7: **end while**

---

$\Phi$ an invariant predictor (although REx does not use such an explicit decomposition). Koyama & Yamaguchi (2020) prove that a **maximal invariant predictor**, that is, one that maximizes mutual information with the targets, $\Phi^* \doteq \mathrm{argmax}_\Phi I(\Phi, Y)$, solves the robust optimization problem (Eqn. 1) under fairly general assumptions; when $\Phi^*$ is unique, we call the features it ignores **spurious**.

The result of Koyama & Yamaguchi (2020) provides a theoretical reason for favoring invariant prediction over the common approach of learning **invariant *representations*** (Pan et al., 2010), which make $P_e(\Phi)$ or $P_e(\Phi|Y)$ equal $\forall e \in \mathcal{E}$. Popular methods here include **Adversarial domain adaptation (ADA)** (Ganin et al., 2016) and **conditional ADA (C-ADA)** (Long et al., 2018). Unlike invariant predictors, invariant representations can easily fail to generalize OOD: ADA forces the predictor to have the same marginal predictions $\hat{P}(Y)$, which is a mistake when $P(Y)$ in fact changes across domains (Zhao et al., 2019); C-ADA suffers from more subtle issues (Arjovsky et al., 2019).

Causal relationships are a paradigmatic example of invariant relationships. In a **Structural Causal Model (SCM)**, the **mechanism** for a variable describes how it's value is computed based on the values of its causes, which can be thought of as its parents in a Pytorch (Paszke et al., 2019) computational graph.[4] Works that take a causal approach to domain generalization often assume that the mechanism for $Y$ is fixed, while $X$ may be subject to different interventions in different domains (Bühlmann, 2018a). We call resulting changes in $P(X, Y)$ **interventional shift**. Interventional shift can involve both covariate shift and/or concept shift. In their seminal work on **Invariant Causal Prediction (ICP)**, Peters et al. (2016) leverage this invariance to learn which elements of $X$ cause $Y$. ICP and its nonlinear extension (Heinze-Deml et al., 2018) use statistical tests to detect whether the residuals of a linear model are equal across domains. Our work differs from ICP in that:

1. Our method is model agnostic and scales to deep networks.

2. Our goal is OOD generalization, not causal inference. These are not identical: invariant prediction can sometimes make use of non-causal relationships, but when deciding which interventions to perform, a truly causal model is called for.

3. Our learning principle only requires invariance of risks, not residuals. Nonetheless, we prove that this can ensure invariant causal prediction.

A more similar method to REx is **Invariant Risk Minimization (IRM)** (Arjovsky et al., 2019), which shares properties (1) and (2) of the list above. Like REx, IRM also uses a weaker form of invariance than ICP; namely, they insist that the optimal linear classifier must match across domains.[5] Still, REx differs significantly from IRM. While IRM specifically aims for invariant prediction, REx seeks robustness to whatever forms of distributional shift are present. Thus, REx is more directly focused on the problem of OOD generalization, and can provide robustness to a wider variety of distributional shifts, inluding covariate shift. Also, unlike REx, IRM seeks to match $\mathbb{E}(Y|\Phi(X))$ across domains, not the full $P(Y|\Phi(X))$. This, combined with IRM's indifference to covariate shift, make it more effective in cases where different domains or examples are inherently more noisy.

### 2.3 FAIRNESS

Equalizing risk across different groups (e.g. male vs. female) has been proposed as a definition of **fairness** (Donini et al., 2018), generalizing the equal opportunity definition of fairness (Hardt et al.,

---

[4]See Appendix for a more in-depth technical overview of SCMs.
[5]In practice, IRMv1 replaces this bilevel optimization problem with a gradient penalty on classifier weights.

2016). Williamson & Menon (2019) propose using the variance or absolute difference of risks to measure deviation from this notion of fairness; these correspond to our V-REx and (in the case of only two domains) MM-REx. However, in the context of fairness, equalizing the risk of training groups is the *goal*. Our work goes beyond this by showing that it can serve as a *method* for OOD generalization.

## 3 RISK EXTRAPOLATION

Before discussing algorithms for REx and theoretical results, we first expand on our high-level explanations of what REx does, what kind of OOD generalization it promotes, and how. The principle of Risk Extrapolation (REx) has two aims:

1. Reducing training risks
2. Increasing similarity of training risks

In general, these goals can be at odds with each other; decreasing the risk in the domain with the lowest risk also decreases the overall similarity of training risks. Thus methods for REx typically seeks to *increase* risk on the best performing domains. From a geometric point of view, encouraging equality of risks flattens the "risk plane" (the affine span of the training risks, considered as a function of the data distribution, see Figure 1). While this can result in higher training risks, it also means that the risk changes less if the distributional shifts between training domains are magnified at test time.

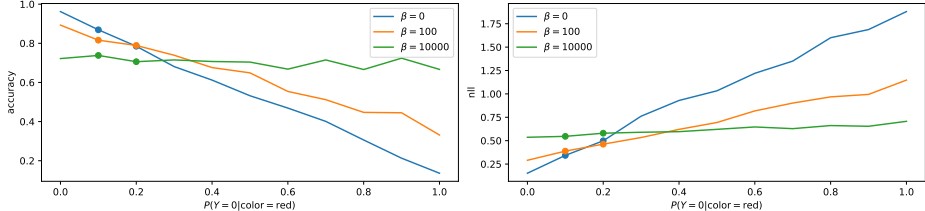

Figure 2: Training accuracies (**left**) and risks (**right**) on colored MNIST domains with varying $P(Y = 0|\text{color} = \text{red})$ after 500 epochs. Dots represent training risks, lines represent test risks. Increasing the V-REx penalty (i.e. $\beta$) leads to a flatter "risk plane" and more consistent performance across domains, as the model learns to ignore color in favor of shape-based invariant prediction.

Figure 2 illustrates how flattening the risk plane can promote OOD generalization on real data, using the Colored MNIST (CMNIST) task as an example (Arjovsky et al., 2019). In the CMNIST training domains, the color of a digit is more predictive of the label than the shape is. But because the correlation between color and label is not invariant, predictors that use the color feature achieve different risk on different domains. By enforcing equality of risks, REx prevents the model from using the color feature enabling successful generalization to the test domain where the correlation between color and label is reversed.

**Probabilities vs. Risks.** Figure 3 depicts how the extrapolated risks considered in MM-REx can be translated into a corresponding change in $P(X, Y)$. Training distributions can be thought of as points in an affine space with a dimension for every possible value of $(X, Y)$; see Appendix C.1 for an example. Because the risk is linear w.r.t. $P(x, y)$, a convex combination of risks from different domains is equivalent to the risk on a domain given by the mixture of their distributions. The same holds for the affine combinations used in MM-REx, with the caveat that the negative coefficients may lead to negative probabilities, making the resulting $P(X, Y)$ a **quasiprobability distribution**, i.e. a signed measure with intregral 1. We explore the theoretical implications of this in Section E

**Covariate Shift.** When only $P(X)$ differs across domains, as in Figure 3, then $\Phi(x) = x$ is an invariant predictor, and thus learning an invariant predictor is not expected to improve OOD generalization (compared with ERM). Instead, what is needed is robustness to covariate shift, which REx, but not IRM, can provide.[6] Robustness to covariate shift can improve OOD generalization by

---

[6]Arjovsky et al. (2019) recognize this limitation of IRM in what they call the "realizable" case.

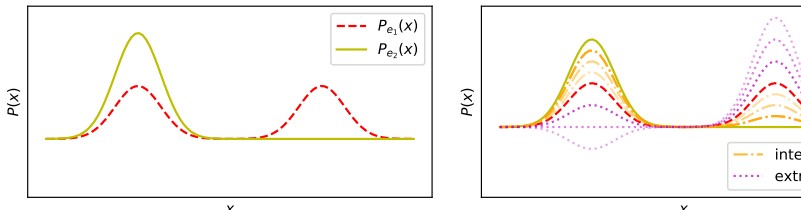

Figure 3: Extrapolation can yield a distribution with *negative* $P(x)$ for some $x$. **Left:** $P(x)$ for domains $e_1$ and $e_2$. **Right:** Point-wise interpolation/extrapolation of $P^{e_1}(x)$ and $P^{e_2}(x)$.

ensuring that low-capacity models spend sufficient capacity on low-density regions of the input space, as we demonstrate in Appendix C.2. But even for high capacity models, $P(X)$ can have a significant influence on what is learned; for instance Sagawa et al. (2019) show that DRO can significantly improves the performance on rare groups in their with a model that achieves 100% training accuracy in their Waterbirds dataset. Pursuing robustness to covariate shift also comes with drawbacks for REx, however: REx does not distinguish between underfitting and inherent noise in the data, and so can force the model to make equally bad predictions everywhere, even if some examples are less noisy than others.

### 3.1 METHODS OF RISK EXTRAPOLATION

We now formally describe the **Minimax REx (MM-REx)** and **Variance-REx (V-REx)** techniques for risk extrapolation. Minimax-REx performs robust learning over a perturbation set of *affine* combinations of training risks with bounded coefficients:

$$\mathcal{R}_{\text{MM-REx}}(\theta) \doteq \max_{\substack{\Sigma_e \lambda_e = 1 \\ \lambda_e \geq \lambda_{\min}}} \sum_{e=1}^{m} \lambda_e \mathcal{R}_e(\theta) = (1 - m\lambda_{\min}) \max_e \mathcal{R}_e(\theta) + \lambda_{\min} \sum_{e=1}^{m} \mathcal{R}_e(\theta), \quad (5)$$

where $m$ is the number of domains, and the hyperparameter $\lambda_{\min}$ controls how much we extrapolate. For negative values of $\lambda_{\min}$, MM-REx places negative weights on the risk of all but the worst-case domain, and as $\lambda_{\min} \to -\infty$, this criterion enforces strict equality between training risks; $\lambda_{\min} = 0$ recovers risk interpolation (RI). Thus, like RI, MM-REx aims to be robust in the direction of variations in $P(X, Y)$ between test domains. However, negative coefficients allow us to extrapolate to more extreme variations. Geometrically, larger values of $\lambda_{\min}$ expand the perturbation set farther away from the convex hull of the training risks, encouraging a flatter "risk-plane" (see Figure 2).

While MM-REx makes the relationship to RI/RO clear, we found using the variance of risks as a regularizer (V-REx) simpler, stabler, and more effective:

$$\mathcal{R}_{\text{V-REx}}(\theta) \doteq \beta \, \text{Var}(\{\mathcal{R}_1(\theta), ..., \mathcal{R}_m(\theta)\}) + \sum_{e=1}^{m} \mathcal{R}_e(\theta) \quad (6)$$

Here $\beta \in [0, \infty)$ controls the balance between reducing average risk and enforcing equality of risks, with $\beta = 0$ recovering ERM, and $\beta \to \infty$ leading V-REx to focus entirely on making the risks equal. See Appendix for the relationship between V-REx and MM-REx and their gradient vector fields.

## 4 EXPERIMENTS

We evaluate REx and compare with IRM on a range of tasks requiring OOD generalization. REx provides generalization benefits and outperforms IRM on a wide range of tasks, including: i) variants of the Colored MNIST (CMNIST) dataset (Arjovsky et al., 2019) with covariate shift, ii) continuous control tasks with partial observability and spurious features, iii) domain generalization tasks from the DomainBed suite (Gulrajani & Lopez-Paz, 2020). On the other hand, when the inherent noise in $Y$ varies across environments, IRM succeeds and REx performs poorly.

### 4.1 COLORED MNIST

Arjovsky et al. (2019) construct a binary classification problem (with 0-4 and 5-9 each collapsed into a single class) based on the MNIST dataset, using color as a spurious feature. Specifically, digits

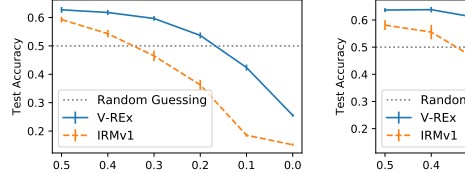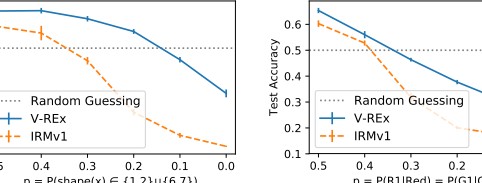

Figure 4: REx outperforms IRM on Colored MNIST variants that include covariate shift. The x-axis indexes increasing amount of shift between training distributions, with $p = 0$ corresponding to disjoint supports. **Left:** class imbalance, **Center:** shape imbalance, **Right:** color imbalance.

are either colored red or green, and there is a strong correlation between color and label, which is reversed at test time. The goal is to learn the causal "digit shape" feature and ignore the anti-causal "digit color" feature. The learner has access to three domains:

1. A training domain where green digits have a 80% chance of belonging to class 1 (digits 5-9).
2. A training domain where green digits have a 90% chance of belonging to class 1.
3. A test domain where green digits have a 10% chance of belonging to class 1.

We use the exact same hyperparameters as Arjovsky et al. (2019), only replacing the IRMv1 penalty with MM-REx or V-REx penalty.[7] These methods all achieve similar performance, see Table 2.

**CMNIST with covariate shift**   To test our hypothesis that REx should outperform IRM under covariate shift, we construct 3 variants of the CMNIST dataset. Each variant represents a different way of inducing covariate shift to ensure differences across methods are consistent. These experiments combine covariate shift with interventional shift, since $P(Green|Y = 1)$ still differs across training domains as in the original CMNIST.

| Method | train acc | test acc |
|---|---|---|
| **V-REx (ours)** | $71.5 \pm 1.0$ | $\mathbf{68.7 \pm 0.9}$ |
| IRM | $70.8 \pm 0.9$ | ~~$66.9 \pm 2.5$~~ |
| MM-REx (ours) | $72.4 \pm 1.8$ | $66.1 \pm 1.5$ |
| RI | $88.9 \pm 0.3$ | ~~$22.3 \pm 4.6$~~ |
| ERM | $87.4 \pm 0.2$ | ~~$17.1 \pm 0.6$~~ |
| Grayscale oracle | $73.5 \pm 0.2$ | ~~$73.0 \pm 0.4$~~ |
| Optimum | 75 | 75 |
| Chance | 50 | 50 |

Table 2:   Accuracy (percent) on Colored MNIST. REx and IRM learn to ignore the spurious color feature. ~~Strikethrough~~ results achieved via tuning on the test set.

1. **Class imbalance:** varying $p = P(\text{shape(x)} \in \{0, 1, 2, 3, 4\})$; as in Wu et al. (2020).
2. **Digit imbalance:** varying $p = P(\text{shape(x)} \in \{1, 2\} \cup \{6, 7\})$; digits 0 and 5 are removed.
3. **Color imbalance:** We use 2 versions of each color, for 4 total channels: $R_1, R_2, G_1, G_2$. We vary $p = P(R_1|Red) = P(G_1|Green)$.

While (1) also induces change in $P(Y)$, (2) and (3) induce *only* covariate shift in the causal shape and anti-causal color features (respectively). We compare across several levels of imbalance, $p \in [0, 0.5]$, using the same hyperparameters from Arjovsky et al. (2019), and plot the mean and standard error over 3 trials.

V-REx significantly outperforms IRM in every case, see Figure 4. In order to verify that these results are not due to bad hyperparameters for IRM, we perform a random search that samples 340 unique hyperparameter combinations for each value of $p$, and compare the the number of times each method achieves better than chance-level (50% accuracy). Again, V-REx outperforms IRM; in particular, for small values of $p$, IRM never achieves better than random chance performance, while REx does better than random in 4.4%/23.7%/2.0% of trials, respectively, in the class/digit/color imbalance scenarios for $p = 0.1/0.1/0.2$. This indicates that REx can achieve good OOD generalization in settings involving both covariate and interventional shift, whereas IRM struggles to.

## 4.2   TOY STRUCTURAL EQUATION MODELS (SEMs)

REx's sensitivity to covariate shift can also be a weakness when reallocating capacity towards domains with higher risk does not help the model reduce their risk, e.g. due to irreducible noise. We illustrate

---

[7]When there are only 2 domains, MM-REx is equivalent to a penalty on the Mean Absolute Error (MAE), see Appendix F.2.2.

| Algorithm | ColoredMNIST | VLCS | PACS | OfficeHome |
|---|---|---|---|---|
| ERM | $52.0 \pm 0.1$ | $77.4 \pm 0.3$ | $85.7 \pm 0.5$ | $67.5 \pm 0.5$ |
| IRM | $51.8 \pm 0.1$ | $78.1 \pm 0.0$ | $84.4 \pm 1.1$ | $66.6 \pm 1.0$ |
| V-REx | $52.1 \pm 0.1$ | $77.9 \pm 0.5$ | $85.8 \pm 0.6$ | $66.7 \pm 0.5$ |

Table 3: REx, IRM, and ERM all perform comparably on set of domain generalization benchmarks.

this using the linear-Gaussian structural equation model (SEM) tasks introduced by Arjovsky et al. (2019). Like CMNIST, these SEMs include spurious features by construction. They also introduce 1) heteroskedasticity, 2) hidden confounders, and/or 3) elements of $X$ that contain a mixture of causes and effects of $Y$. These three properties highlight advantages of IRM over ICP (Peters et al., 2016), as demonstrated empirically by Arjovsky et al. (2019). REx is also able to handle (2) and (3), but it performs poorly in the heteroskedastic tasks. See Appendix G.2 for details and Table 4 for results.

### 4.3 DOMAIN GENERALIZATION IN THE DOMAINBED SUITE

Methodologically, it is inappropriate to assume access to the test environment in domain generalization settings, as the goal is to find methods which generalize to *unknown* test distributions. Gulrajani & Lopez-Paz (2020) introduced the DomainBed evaluation suite to rigorously compare existing approaches to domain generalization, and found that no method reliably outperformed ERM. We evaluate V-REx on DomainBed using the most commonly used training-domain validation set method for model selection. Due to limited computational resources, we limited ourselves to the 4 cheapest datasets. Results of baseline are taken from Gulrajani & Lopez-Paz (2020), who compare with more methods. Results in Table 3 give the average over 3 different train/valid splits.

### 4.4 REINFORCEMENT LEARNING WITH PARTIAL OBSERVABILITY AND SPURIOUS FEATURES

Finally, we turn to reinforcement learning, where covariate shift (potentially favoring REx) and heteroskedasticity (favoring IRM) both occur naturally as a result of randomness in the environment and policy. In order to show the benefits of invariant prediction, we modify tasks from the Deepmind Control Suite (Tassa et al., 2018) to include spurious features in the observation, and train a Soft Actor-Critic (Haarnoja et al., 2018) agent. REx outperforms both IRM and ERM, suggesting that REx's robustness to covariate shift outweighs the challenges it faces with heteroskedasticity in this setting, see Figure 5. We average over 10 runs on `finger_spin` and `walker_walk`, using hyperparameters tuned on `cartpole_swingup` (to avoid overfitting). See Appendix for details and further results.

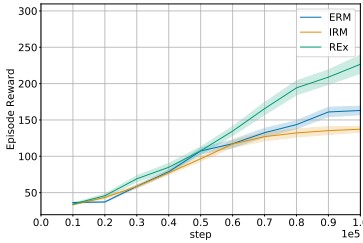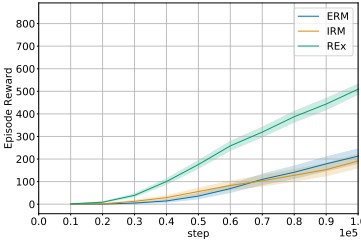

Figure 5: Performance and standard error on `walker_walk` (**left**), `finger_spin` (**right**).

## 5 CONCLUSION

We have demonstrated that REx, a method for robust optimization, can provide robustness and hence out-of-distribution generalization in the challenging case where $X$ contains both causes and effects of $Y$. In particular, like IRM, REx can perform causal identification, but REx can also perform more robustly in the presence of covariate shift. Covariate shift is known to be problematic when models are misspecified, when training data is limited, or does not cover areas of the test distribution. As such situations are inevitable in practice, REx's ability to outperform IRM in scenarios involving a combination of covariate shift and interventional shift makes it a powerful approach.

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

# Appendices

## A    APPENDIX OVERVIEW

Our code is available online at: https://anonymous.4open.science/r/12747e81-8505-43cb-b54e-e75e2344a397/.

The sections of our appendix are as follows:

A) Overview

B) Definition and discussion of extrapolation in machine learning

C) Illustrative examples of how REx works in toy settings

D) Background on causal models

E) Theory

F) The relationship between MM-REx vs. V-REx, and the role each plays in our work

G) Further results and details for experiments mentioned in main text

H) Experiments not mentioned in main text

I) Overview of other topics related to OOD generalization

## B    DEFINITION AND DISCUSSION OF EXTRAPOLATION IN MACHINE LEARNING

We define interpolation and extrapolation as follows: **interpolation** refers to making decisions or predictions about points *within* the convex hull of the training examples and **extrapolation** refers to making decisions or predictions about points *outside* their convex hull.[8] This generalizes the familiar sense of these terms for one-dimensional functions. An interesting consequence of this definition is: for data of high intrinsic dimension, generalization *requires* extrapolation (Hastie et al., 2009), even in the i.i.d. setting. This is because the volume of high-dimensional manifolds concentrates near their boundary; see Figure 6.

**Extrapolation in the space of risk functions.**    The same geometric considerations apply to extrapolating to new domains. Domains can be highly diverse, varying according to high dimensional attributes, and thus requiring extrapolation to generalize across. Thus Risk Extrapolation might often do a better job of including possible test domains in its perturbation set than Risk Interpolation does.

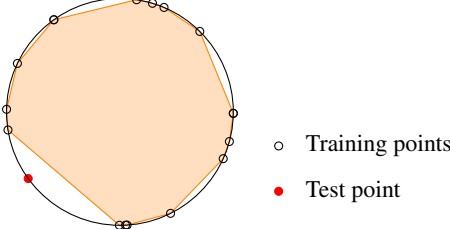

○  Training points

●  Test point

Figure 6: Illustration of the importance of extrapolation for generalizing in high dimensional space. In high dimensional spaces, mass concentrates near the boundary of objects. For instance, the uniform distribution over a ball in $N + 1$-dimensional space can be approximated by the uniform distribution over the $N$-dimensional hypersphere. We illustrate this in 2 dimensions, using the 1-sphere (i.e. the unit circle). Dots represent a finite training sample, and the shaded region represents the convex hull of all but one member of the sample. Even in 2 dimensions, we can see why any point from a finite sample from such a distribution remains outside the convex hull of the other samples, with probability 1. The only exception would be if two points in the sample coincide *exactly*.

---

[8]Surprisingly, we were not able to find any existing definition of these terms in the machine learning literature. They have been used in this sense (Hastie et al., 2009; Haffner, 2002), but also to refer to strong generalization capabilities more generally (Sahoo et al., 2018).

## C ILLUSTRATIVE EXAMPLES OF HOW REx WORKS IN TOY SETTINGS

Here, we work through two examples to illustrate:

1. How to understand extrapolation in the space of probability density/mass functions (PDF/PMFs)

2. How REx encourages robustness to covariate shift via distributing capacity more evenly across possible input distributions.

### C.1 6D EXAMPLE OF REx

Here we provide a simple example illustrating how to understand extrapolations of probability distributions. Suppose $X \in \{0, 1, 2\}$ and $Y \in \{0, 1\}$, so there are a total of 6 possible types of examples, and we can represent their distributions in a particular domain as a point in 6D space: $(P(0, 0), P(0, 1), P(1, 0), P(1, 1), P(2, 0), P(2, 1))$. Now, consider three domains $e_1, e_2, e_3$ given by

1. $(a, b, c, d, e, f)$

2. $(a, b, c, d, e - k, f + k)$

3. $(2a, 2b, c(1 - \frac{a+b}{c+d}), d(1 - \frac{a+b}{c+d}), e, f)$

The difference between $e_1$ and $e_2$ corresponds to a shift in $P(Y|X = 2)$, and suggests that $Y$ cannot be reliably predicted across different domains when $X = 2$. Meanwhile, the difference between $e_1$ and $e_3$ tells us that the relative probability of $X = 0$ vs. $X = 1$ can change, and so we might want our model to be robust to these sorts of covariate shifts. Extrapolating risks across these 3 domains effectively tells the model: "don't bother trying to predict $Y$ when $X = 2$ (i.e. aim for $\hat{P}(Y = 1|X = 2) = .5$), and split your capacity equally across the $X = 0$ and $X = 1$ cases". By way of comparison, IRM would also aim for $\hat{P}(Y = 1|X = 2) = .5$, whereas ERM would aim for $\hat{P}(Y = 1|X = 2) = \frac{3f+k}{3e+3f}$ (assuming $|D_1| = |D_2| = |D_3|$). And unlike REx, both ERM and IRM would split capacity between $X = 0/1/2$ cases according to their empirical frequencies.

### C.2 COVARIATE SHIFT EXAMPLE

Viewing REx as robust learning over the affine span of the training distributions reveals its potential to improve robustness to distribution shifts. Consider a situation in which a model encounters two types of inputs: COSTLY inputs with probability $q$ and CHEAP inputs with probability $1 - q$. The model tries to predicts the input – it outputs COSTLY with probability $p$ and CHEAP with probability $1 - p$. If the model predicts right its risk is 0, but if it predicts COSTLY instead of CHEAP it gets a risk $u = 2$, and if it predicts CHEAP instead of COSTLY it gets a risk $v = 4$. The risk has expectation $\mathcal{R}_q(p) = (1 - p)(1 - q)u + pqv$. We have access to two domains with different input probabilities $q_1 < q_2$. This is an example of pure covariate shift.

We want to guarantee the minimal risk over the set of all possible domains:

$$\min_{p \in [0,1]} \max_{q \in [0,1]} \mathcal{R}_q(p) = (1 - p)(1 - q)u + pqv$$

as illustrated in Figure 7. The saddle point solution of this problem is $p = \omega = {}^u\!/_{u+v}$ and $\mathcal{R}_q(p) = {}^{uv}\!/_{u+v}, \forall q$. From the figure we see that $\mathcal{R}_{q_1}(p) = \mathcal{R}_{q_2}(p)$ can only happen for $p = \omega$, so the risk extrapolation principle will return the minimax optimal solution.

If we use ERM to minimize the risk, we will pool together the domains into a new domain with COSTLY input probability $\bar{q} = (q_1 + q_2)/2$. ERM will return $p = 0$ if $\bar{q} > \omega$ and $p = 1$ otherwise. Risk interpolation (RI) $\min_p \max_{q \in \{q_1, q_2\}} R_q(p)$ will predict $p = 0$ if $q_1, q_2 > \omega$, $p = 1$ if $q_1, q_2 < \omega$ and $p = \omega$ if $q_1 < \omega < q_2$. We see that only REx finds the minimax optimum for arbitrary values of $q_1$ and $q_2$.

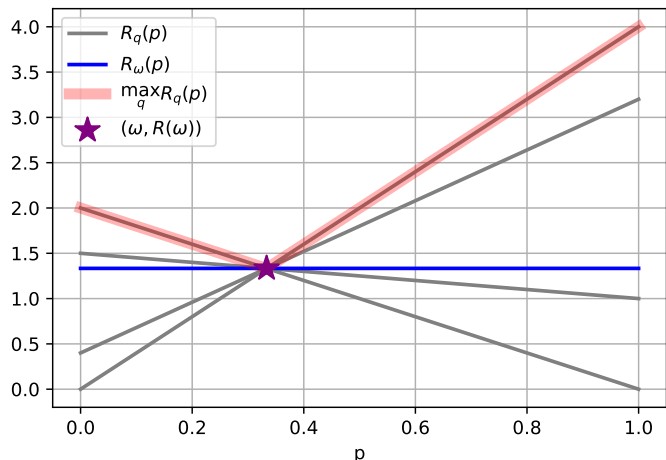

Figure 7: Each grey line is a risk $\mathcal{R}_q(p)$ as functions of $p$ for a specific value of $q$. The blue line is when $q = \omega$. We highlight in red the curve $\max_q \mathcal{R}_q(p)$ whose minimum is the saddle point marked by a purple star in $p = \omega$.

## D    BACKGROUND ON CAUSAL MODELS

### D.1    CAUSAL MODELS: DEFINITIONS AND NOTATION

A **causal graphical model** (Koller & Friedman, 2009; Pearl, 2009) is a directed acyclic graph, where edges point from causes (including noise variables) to effects. Specifying the way an effect $N$ is computed based on the values of its causes (the **mechanism**, **structural equation**, or **structural assignment** for $Y$, commonly denoted $f_N$, although we use $m^N$ in this work) yields a **Structural Causal Model (SCM)**, denoted by $\mathfrak{C}$. We refer the reader to Peters et al. (2017), definition 6.2, for more technical details.[9] At a high level, SCMs are to causal graphical models as joint probability distributions are to probabilistic graphical models (PGMs); rather than just specifying causal or independence assumption, they fill in the details of what values each node can take on and how exactly the nodes influence or depend on one another.

To highlight the difference between causal and probabilistic models, it helps to think of an SCM as analogous to a computational graph in a deep learning framework such as Pytorch or TensorFlow, with root nodes corresponding to independent noise variables. Elaborating: both objects specify how to compute a node, given the values of its parents.[10] And while the value of a node may be *statistically* dependent on any node to which it is path-connected, its value is only *directly* influenced by the value of its parents, or by the programmer setting it to a particular value. In this analogy, setting the value of the node corresponds to an *intervention*. More formally, an **intervention**, $\iota$ can be any modification to the mechanisms of one or more variables; it can even introduce new edges, so long as it does not introduce a cycle. Applying an intervention on $\mathfrak{C}$ defines a new SCM, $\mathfrak{C}^\iota$.

The mechanisms of $\mathfrak{C}$ describe a generative process for the data: absent any conditioning or interventions (which are viewed as *producing a new SCM*, rather than *modifying* $\mathfrak{C}$), the data can be generated using ancestral sampling, defining a joint distribution over all the nodes. This sampling process where, $N = m^N(Pa(N), \varepsilon^N)$, defines an **entailed distribution**, $P_N^{\mathfrak{C}}(N)$ for every node $N$ in $\mathfrak{C}$. Examining this sampling process reveals that the entailed distribution of $N$ depends only on its mechanism and the value of its parents. As a simple consequence, we have that: $P_Y^{\mathfrak{C}}(Y|Q) = P(Y|Pa(Y)) = P(Y|do(Q = q))$ for $Q$ representing all the other nodes in $\mathfrak{C}$. Al-

---

[9]SCMs are sometimes called **Structural Equation Models (SEMs)**, although this term is often reserved for SCMs with linear mechanisms. We do not make assumptions about SCM mechanisms (e.g. linearity), unless stated otherwise.

[10]In an SCM, every node has its own unique independent noise variable, $\varepsilon^N$, as a parent, which is independent of all other nodes. Any "inherent randomness" in $N$ arises from $\varepsilon^N$.

though all the nodes of $\mathfrak{C}$ will not necessarily be observed, we assume that they are (i.e. $X = Q$ for supervised learning) for the remainder of the appendix, and overload $m^Y(x)$ to refer to the entailed distribution for $Y$, given the values of its parents (only).

## D.2 Causal approaches to domain generalization

A number of works draw connections between causality, domain generalization, and robustness. Storkey (2009); Moreno-Torres et al. (2012); Schölkopf et al. (2012) characterize different types of distributional shift in probabilistic and causal terms. A common thread in these works is that the statistical relationship $P(\text{effect}|\text{cause})$ should typically be preserved. In other words, interventions are assumed to be rare or sparse; or perhaps only certain types of interventions are allowed. For instance, the assumption of constant $P(Y|X)$ is appropriate when $X$ causes $Y$, and $X$ but not $Y$ may be subject to interventions. But when the roles are reversed, and $Y$ causes $X$, constant $P(X|Y)$ is a better assumption (Lipton et al., 2018). Bühlmann (2018b) and Rothenhäusler et al. (2018) describe causal learning as a special case of robust learning, with the perturbation set given by the set of possible interventions that do not change the mechanism of $Y$.

## E    Theory

### E.1    Causal discovery with REx

We prove that Risk Extrapolation learns the causal mechanism of $Y$ under the same assumptions as used by Peters et al. (2016) in their work on Invariant Causal Prediction (ICP). These are somewhat restrictive:[11]

- We assume that the causes of $Y$ are observed, i.e. $Pa(Y) \subseteq X$.

- We assume that the homoskedasticity (a slight generalization of the additive noise setting assumed by Peters et al. (2016)).

The contribution of our theory (vs. ICP) is to prove that equalizing risks is sufficient to learn the causes of $Y$. In contrast, they insist that the entire distribution of error residuals (in predicting $Y$) be the same across domains.

Theorem 1 demonstrates a practical result: we only need 3 interventions on each dimension of the input in order to identify a linear SEM model using REx. Theorem 2 on the other hand, is meant to provide insight into how the REx principle relates to causal invariance, not a practical result; the perturbation set in these theorems is uncountably infinite. It might also be possible to prove that REx identifies underlying causal structure in the limit of countably many diverse interventions, given some assumptions about the underlying casual structure, but we leave this to future work.

### E.1.1    The linear case

We begin with a theorem based on the setting explored by Peters et al. (2016):

**Theorem 1.** *Given a Linear SEM, $X_i \leftarrow \sum_{j \neq i} \beta_{(i,j)} X_j + \varepsilon_i$, with $Y \doteq X_0$, suppose a predictor $f_\beta(X) = \sum_{j:j>0} \beta_j X_j + \varepsilon_j$ satisfies REx for the mean-squared error (MSE) and a perturbation set of domains that contains 3 distinct $do()$ interventions for each $X_i : i > 0$. Then $\beta_j = \beta_0, j$, for all $j$.*

*Proof.* We adapt the proof of Theorem 4i from Peters et al. (2016) to show that REx will learn the correct model under similar assumptions. Let $Y \leftarrow \gamma X + \varepsilon$ be the mechanism for $Y$, assumed to be fixed across all domains, and let $\hat{Y} = \beta X$ be our predictor. Then the residual is $R(\beta) = (\gamma - \beta)X + \varepsilon$. Define $\alpha_i \doteq \gamma_i - \beta_i$, and consider an intervention $do(X_j = x)$ on the youngest node $X_j$ with $\alpha_j \neq 0$. Then as in eqn 36/37 of Peters et al. (2016), we compare the residuals $R$ of this intervention and of the observational distribution:

---

[11]Although we believe they could be substantially weakened, we emphasize that the goal of REx is OOD generalization, not causal discovery, which we view as merely a means to that end. Thus, in cases where the causal model is not a maximal invariant predictor (MIP), we expect REx to learn the MIP, *not* the mechanism.

$$R^{\text{obs}}(\beta) = \alpha_j X_j + \sum_{i \neq j} \alpha_i X_i + \varepsilon \qquad R^{do(X_j=x)}(\beta) = \alpha_j x + \sum_{i \neq j} \alpha_i X_i + \varepsilon \qquad (7)$$

We now compute the MSE risk for both domains, set them equal, and simplify to find a quadratic formula for $x$:

$$\mathbb{E}\left[ \left( \alpha_j X_j + \sum_{i \neq j} \alpha_i X_i + \varepsilon \right)^2 \right] = \mathbb{E}\left[ \left( \alpha_j x + \sum_{i \neq j} \alpha_i X_i + \varepsilon \right)^2 \right] \qquad (8)$$

$$0 = \alpha_j^2 x^2 + 2\alpha_j \mathbb{E}[\sum_{i \neq j} \alpha_i X_i + \varepsilon]x - \mathbb{E}\left[ (\alpha_j X_j)^2 - 2\alpha_j X_j (\sum_{i \neq j} \alpha_i X_i + \varepsilon) \right] \qquad (9)$$

Since there are at most two values of $x$ that satisfy this equation, any other value leads to a violation of REx, so that $\alpha_j$ needs to be zero – contradiction. In particular having domains with 3 different $do$-interventions on every $X_i$ guarantees that the risks are not equal across all domains. $\qquad\square$

### E.1.2 THE GENERAL CASE

Given the assumption that a predictor satisfies REx over *all* interventions that do not change the mechanism of $Y$, we can prove a much more general result.

**Preliminaries:** Our setting and assumptions for this case are:

1. We consider an arbitrary SCM, $\mathfrak{C}$, generating $Y$ and $X$.
2. We use $m^Y$ to denote the mechanism for $Y$, as well as the entailed distribution $m^Y(x) \doteq P_Y^{\mathfrak{C}}(Y) = P(Y|Pa(Y)) = P(Y|do(X = x))$.
3. An intervention $\iota$ represents an arbitrary change to the mechanisms of $\mathfrak{C}$, similarly to Arjovsky et al. (2019), and we define the **causal perturbation set of $Y$** $\mathcal{E}^I$ as the set of domains whose intervention does not change $m^Y$, similarly to Peters et al. (2016).
4. We assume that $\ell$ is a (strictly) proper scoring rule.
5. We say the SEM is **homoskedastic** when the Bayes error rate for $\ell(m^Y(x), m^Y(x))$ is the same for all $x \in \mathcal{X}$, otherwise, it is **heteroskedastic**.[12]

**Theorem 2.** *In the homoskedastic case, a predictor that satisfies REx over $\mathcal{E}^I$ uses $m^Y(x)$ as its predictive distribution on input $x$ for all $x \in \mathcal{X}$.*

The intuition of the proof is as follows:

1. We first show that that the distribution of $Y$ given its parents doesn't depend on the domain, and so $m^Y$ can be used to make reliable point-wise predictions across domains.
2. This translates into equality of risk across domains when the overall difficulty of the examples is held constant across domains, by assuming homoskedasticity.[13]
3. While a different predictor might do a better job on some domains, we can always find an domain where it does worse than $m^Y$, and so $m^Y$ is both unique and optimal.

We emphasize that the predictor is not restricted to any particular class of models, and is a generic function $f : \mathcal{X} \to \mathcal{P}(Y)$, where $\mathcal{P}(Y)$ is the set of distributions over $Y$. Hence, we drop $\theta$ from the below discussion and simply use $f$ to represent the predictor, and $\mathcal{R}(f)$ its risk.

---

[12] Note that our definitions of **homoskedastic/heteroskedastic** do *not* correspond to the types of domains constructed in Arjovsky et al. (2019), Section 5.1, but rather are a generalization of the definitions of these terms as commonly used in statistics. Specifically, for us, *hetero*skedasticity means that the "predicatability" (e.g. variance) of $Y$ differs across inputs $x$, whereas for Arjovsky et al. (2019), it means the predicatability of $Y$ at a given input varies across *domains*; we refer to this second type as *domain*-homo/heteroskedasticity for clarity.

[13] Note we could also assume no covariate shift in order to fix the difficulty, but this seems hard to motivate in the context of interventions on $X$.

*Proof.* Let $\mathcal{R}^e(f, x)$ be the loss of predictor $f$ on point $x$ in domain $e$, and $\mathcal{R}^e(f) = \int_{P^e(x)} \mathcal{R}^e(f, x)$ be the risk of $f$ in $e$. Define $\iota(x)$ as the domain given by the intervention $do(X = x)$, and note that $\mathcal{R}^{\iota(x)}(f) = \mathcal{R}^{\iota(x)}(f, x)$.

Let $X_1 \doteq Pa(Y)$. For every $x \in \mathcal{X}$, $m^Y(x) \doteq P(Y|do(X = x)) = P(Y|do(X_1 = x_1)) = P(Y|X_1 = x_1)$ is invariant (meaning "independent of domain") by definition; $P(Y|do(X = x)) = P(Y|do(X_1 = x_1)) = P(Y|X_1 = x_1)$ follows from the semantics of SEM/SCMs, and the fact that we don't allow $m^Y$ to change across domains. Specifically $Y$ is always generated by the same ancestral sampling process that only depends on $X_1$ and $\varepsilon^Y$. Thus the risk of the predictor $m^Y(x)$ at point $x$, $\mathcal{R}^e(m^Y, x) = \ell(m^Y(x), m^Y(x))$ is also invariant, soit $\mathcal{R}(m^Y, x)$. Thus $\mathcal{R}^e(m^Y) = \int_{P^e(x)} \mathcal{R}^e(m^Y, x) = \int_{P^e(x)} \mathcal{R}(m^Y, x)$ is invariant whenever $\mathcal{R}(m^Y, x)$ does not depend on $x$ (homoskedastic case).

Now, we show that any other $g$ achieves higher risk than $m^Y$ for at least one domain. This demonstrates both that $m^Y$ achieves minimal risk (thus satisfying REx), and that it is the unique predictor which does so (and thus no other predictors satisfy REx). We suppose such a $g$ exists and construct an domain where it achieves higher risk than $m^Y$. Specifically, if $g \neq m^Y$ then let $x \in \mathcal{X}$ be a point such that $g(x) \neq m^Y(x)$. And since $\ell$ is a strictly proper scoring rule, this implies that $\ell(g(x), m^Y(x)) > \ell(m^Y(x), m^Y(x))$. But the is exactly the risk of $g$ on the domain $\iota \doteq \iota(do(X = x))$, and thus $g$ achieves higher risk than $m^Y$ in $\iota$. $\square$

**Remark.** Even given infinite data from a distribution with full support over $\mathcal{E}^I$, the ERM principle does not, in general, recover the causal mechanism for $Y$. Rather, the ERM solution depends on the distribution over domains. As an example, consider the colored MNIST task with $1 - \epsilon$ mass on the training domain where $P(color = Red|Y = 1) = P(color = Green|Y = 0) = .9, P(color = Red|Y = 0) = P(color = Green|Y = 1) = .1, P(Y = 0) = P(Y = 1) = .5$, and $\epsilon$ mass distributed across other domains using a distribution with full support. For $\epsilon$ close to 1, the ERM solution will be dominated by this domain, and the model's predictions will approximate $\hat{P}(Y = 1|color = Red) \approx .9$, $\hat{P}(Y = 1|color = Green) \approx .1$.

### E.2    REx as DRO

We note that MM-REx is also performing robust optimization over a convex hull, see Figure 1. The corners of this convex hull correspond to "extrapolated domains" with coefficients $(\lambda_{\min}, \lambda_{\min}, ..., (1 - (m - 1)\lambda_{\min}))$ (up to some permutation). However, these domains do not necessarily correspond to valid probability distributions; in general, they are quasidistributions, which can assign negative probabilities to some examples. This means that, even if the original risk functions were convex, the extrapolated risks need not be. However, in the case where they *are* convex, then existing theorems, such as the convergence rate result of (Sagawa et al., 2019). This raises several important questions:

1. When is the affine combination of risks convex?
2. What are the effects of negative probabilities on the optimization problem REx faces, and the solutions ultimately found?

We hypothesize that affine combinations of risks will remain convex in mean-squared error regression problems under fairly weak assumptions.

**Negative probabilities:** Figure 8 illustrates this for a case where $\mathcal{X} = \mathbb{Z}_2^2$, i.e. $x$ is a binary vector of length 2. Suppose $x_1, x_2$ are independent in our training domains, and represent the distribution for a particular domain by the point $(P(X_1 = 1), P(X_2 = 1))$. And suppose our 4 training distributions have $(P(X_1 = 1), P(X_2 = 1))$ equal to $\{(.4, .1), (.4, .9), (.6, .1), (.6, .9)\}$, with $P(Y|X)$ fixed.

## F    THE RELATIONSHIP BETWEEN MM-REx VS. V-REx, AND THE ROLE EACH PLAYS IN OUR WORK

The MM-REx and V-REx methods play different roles in our work:

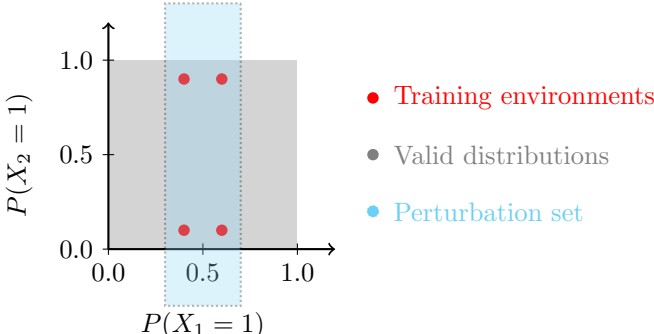

Figure 8: The perturbation set for MM-REx can include "distributions" which assign invalid (e.g. negative) probabilities to some data-points. The range of valid distributions $P(X)$ is shown in grey, and $P(X)$ for 4 different training domains are shown as red points. The interior of the dashed line shows the perturbation set for $\lambda_{\min} = -1/2$.

- We use MM-REx to illustrate that REx can be instantiated as a variant of robust optimization, specifically a generalization of the common Risk Interpolation approach. We also find MM-REx provides a useful geometric intuition, since we can visualize its perturbation set as an expansion of the convex hull of the training risks or distributions.

- We expect V-REx to be the more practical algorithm. It is simple to implement. And it performed better in our CMNIST experiments; we believe this may be due to V-REx providing a smoother gradient vector field, and thus more stable optimization, see Figure F.

Either method recovers the REx principle as a limiting case, as we prove in Section F.1. We also provide a sequence of mathematical derivations that sheds light on the relationship between MM-REx and V-REx in Section F.2 we can view these as a progression of steps for moving from the robust optimization formulation of MM-REx to the penalty term of V-REx:

1. **From minimax to closed form:** We show how to arrive at the closed-form version of MM-REx provided in Eqn. 5.

2. **Closed form as mean absolute error:** The closed form of MM-REx is equivalent to a mean absolute error (MAE) penalty term when there are only two training domains.

3. **V-REx as mean squared error:** V-REx is exactly equivalent to a mean squared error penalty term (always). Thus in the case of only two training domains, the difference between MM-REx and V-REx is just a different choice of norm.

### F.1 V-REX AND MM-REX ENFORCE THE REX PRINCIPLE IN THE LIMIT

We prove that both MM-REx and V-REx recover the constraint of perfect equality between risks in the limit of $\lambda_{\min} \to -\infty$ or $\beta \to \infty$, respectively. For both proofs, we assume all training risks are finite.

**Proposition 1.** *The MM-REx risk of predictor $f_\theta$, $\mathcal{R}_{\mathrm{MM-REx}}(\theta) \to \infty$ as $\lambda_{\min} \to -\infty$ unless $\mathcal{R}^d = \mathcal{R}^e$ for all training domains $d, e$.*

*Proof.* Suppose the risk is not equal across domains, and let the largest difference between any two training risks be $\epsilon > 0$. Then $\mathcal{R}_{\mathrm{MM-REx}}(\theta) = (1 - m\lambda_{\min}) \max_e \mathcal{R}_e(\theta) + \lambda_{\min} \sum_{i=1}^m \mathcal{R}_i(\theta) = \max_e \mathcal{R}_e(\theta) - m\lambda_{\min} \max_e \mathcal{R}_e(\theta) + \lambda_{\min} \sum_{i=1}^m \mathcal{R}_i(\theta) \geq \max_e \mathcal{R}_e(\theta) - \lambda_{\min}\epsilon$, with the inequality resulting from matching up the $m$ copies of $\lambda_{\min} \max_e \mathcal{R}_e$ with the terms in the sum and noticing that each pair has a non-negative value (since $\mathcal{R}_i - \max_e \mathcal{R}_e$ is non-positive and $\lambda_{\min}$ is negative), and at least one pair has the value $-\lambda_{\min}\epsilon$. Thus sending $\lambda \to -\infty$ sends this lower bound on $\mathcal{R}_{\mathrm{MM-REx}}$ to $\infty$ and hence $\mathcal{R}_{\mathrm{MM-REx}} \to \infty$ as well. $\square$

**Proposition 2.** *The V-REx risk of predictor $f_\theta$, $\mathcal{R}_{\mathrm{V-REx}}(\theta) \to \infty$ as $\beta \to \infty$ unless $\mathcal{R}^d = \mathcal{R}^e$ for all training domains $d, e$.*

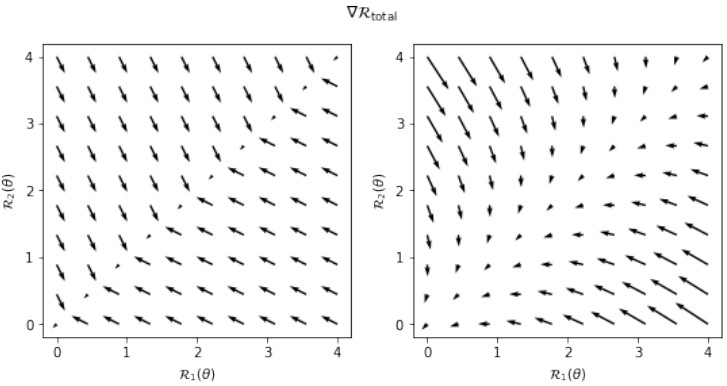

Figure 9: Vector fields of the gradient evaluated at different values of training risks $\mathcal{R}_1(\theta)$, $\mathcal{R}_2(\theta)$. We compare the gradients for $\mathcal{R}_{\text{MM-REx}}$ (**left**) and $\mathcal{R}_{\text{V-REx}}$ (**right**). Note that for $\mathcal{R}_{\text{V-REx}}$, the gradient vectors curve smoothly towards the direction of the origin, as they approach the diagonal (where training risks are equal); this leads to a smoother optimization landscape.

*Proof.* Again, let $\epsilon > 0$ be the largest difference in training risks, and let $\mu$ be the mean of the training risks. Then there must exist an $e$ such that $|\mathcal{R}_e - \mu| \geq \epsilon/2$. And thus $Var_i(\mathcal{R}_i(\theta)) = \sum_i (\mathcal{R}_i - \mu)^2 \geq (\epsilon/2)^2$, since all other terms in the sum are non-negative. Since $\epsilon > 0$ by assumption, the penalty term is positive and thus $\mathcal{R}_{V-REx}(\theta) \doteq \sum_i \mathcal{R}_i(\theta) + \beta Var_i(\mathcal{R}_i(\theta))$ goes to infinity as $\beta \to \infty$. $\qquad\square$

## F.2  CONNECTING MM-REX TO V-REX

### F.2.1  CLOSED FORM SOLUTIONS TO RISK INTERPOLATION AND MINIMAX-REX

Here, we show that risk interpolation is equivalent to the robust optimization objective of Eqn. 4. Without loss of generality, let $\mathcal{R}_1$ be the largest risk, so $\mathcal{R}_e \leq \mathcal{R}_1$, for all $e$. Thus we can express $\mathcal{R}_e = \mathcal{R}_1 - d_e$ for some non-negative $d_e$, with $d_1 = 0 \geq d_e$ for all $e$. And thus we can write the weighted sum of Eqn. 5 as:

$$\mathcal{R}_{\text{MM}}(\theta) \doteq \max_{\substack{\Sigma_e \lambda_e = 1 \\ \lambda_e \geq \lambda_{\min}}} \sum_{e=1}^{m} \lambda_e \mathcal{R}_e(\theta) \tag{10}$$

$$= \max_{\substack{\Sigma_e \lambda_e = 1 \\ \lambda_e \geq \lambda_{\min}}} \sum_{e=1}^{m} \lambda_e (\mathcal{R}_1(\theta) - d_e) \tag{11}$$

$$= \mathcal{R}_1(\theta) + \max_{\substack{\Sigma_e \lambda_e = 2 \\ \lambda_e \geq \lambda_{\min}}} \sum_{e=1}^{m} -\lambda_e(d_e) \tag{12}$$

$$\tag{13}$$

Now, since $d_e$ are non-negative, $-d_e$ is non-positive, and the maximal value of this sum is achieved when $\lambda_e = \lambda_{\min}$ for all $e \geq 2$, which also implies that $\lambda_1 = 1 - (m-1)\lambda_{\min}$. This yields the closed form solution provided in Eqn. 5. The special case of Risk Interpolation, where $\lambda_{\min} = 0$, yields Eqn. 4.

### F.2.2  MINIMAX-REX AND MEAN ABSOLUTE ERROR REX

In the case of only two training risks, MM-REx is equivalent to using a penalty on the mean absolute error (MAE) between training risks. However, penalizing the pairwise absolute errors is not equivalent when there are $m > 2$ training risks, as we show below. Without loss of generality, assume that $\mathcal{R}_1 < \mathcal{R}_2 < ... < \mathcal{R}_m$. Then (1/2 of) the $\mathcal{R}_{\text{MAE}}$ penalty term is:

$$\sum_i \sum_{j \leq i} (\mathcal{R}_i - \mathcal{R}_j) = m\mathcal{R}_m - \sum_{j \leq m} \mathcal{R}_j + (m-1)\mathcal{R}_{m-1} - \sum_{j \leq m-1} \mathcal{R}_j \dots \tag{14}$$

$$= \sum_j j\mathcal{R}_j - \sum_j \sum_{i \leq j} \mathcal{R}_i \tag{15}$$

$$= \sum_j j\mathcal{R}_j - \sum_j (m - j + 1)\mathcal{R}_j \tag{16}$$

$$= \sum_j (2j - m - 1)\mathcal{R}_j \tag{17}$$

For $m = 2$, we have $1/2\mathcal{R}_{\text{MAE}} = (2*1 - 2 - 1)\mathcal{R}_1 + (2*2 - 2 - 1)\mathcal{R}_2 = \mathcal{R}_2 - \mathcal{R}_1$. Now, adding this penalty term with some coefficient $\beta_{\text{MAE}}$ to the ERM term yields:

$$\mathcal{R}_{\text{MAE}} \doteq \mathcal{R}_1 + \mathcal{R}_2 + \beta_{\text{MAE}}(\mathcal{R}_2 - \mathcal{R}_1) = (1 - \beta_{\text{MAE}})\mathcal{R}_1 + (1 + \beta_{\text{MAE}})\mathcal{R}_2 \tag{18}$$
$$\tag{19}$$

We wish to show that this is equal to $\mathcal{R}_{\text{MM}}$ for an appropriate choice of learning rate $\gamma_{\text{MAE}}$ and hyperparameter $\beta_{\text{MAE}}$. Still assuming that $\mathcal{R}_1 < \mathcal{R}_2$, we have that:

$$\mathcal{R}_{\text{MM}} \doteq (1 - \lambda_{\min})\mathcal{R}_2 + \lambda_{\min}\mathcal{R}_1 \tag{20}$$

Choosing $\gamma_{\text{MAE}} = 1/2\gamma_{\text{MM}}$ is equivalent to multiplying $\mathcal{R}_{\text{MM}}$ by 2, yielding:

$$2\mathcal{R}_{\text{MM}} \doteq 2(1 - \lambda_{\min})\mathcal{R}_2 + 2\lambda_{\min}\mathcal{R}_1 \tag{21}$$

Now, in order for $\mathcal{R}_{\text{MAE}} = 2\mathcal{R}_{\text{MM}}$, we need that:

$$2 - 2\lambda_{\min} = 1 + \beta_{\text{MAE}} \tag{22}$$
$$2\lambda_{\min} = 1 - \beta_{\text{MAE}} \tag{23}$$
$$\tag{24}$$

And this holds whenever $\beta_{\text{MAE}} = 1 - 2\lambda_{\min}$. When $m > 2$, however, these are not equivalent, since $R_{\text{MM}}$ puts equal weight on all but the highest risk, whereas $\mathcal{R}_{\text{MAE}}$ assigns a different weight to each risk.

### F.2.3 Penalizing pairwise mean squared error (MSE) yields V-REx

The V-REx penalty (Eqn. 6) is equivalent to the average pairwise mean squared error between all training risks (up to a constant factor of 2). Recall that $\mathcal{R}_i$ denotes the risk on domain $i$. We have:

$$\frac{1}{2n^2} \sum_i \sum_j (\mathcal{R}_i - \mathcal{R}_j)^2 = \frac{1}{2n^2} \sum_i \sum_j (\mathcal{R}_i^2 + \mathcal{R}_j^2 - 2\mathcal{R}_i\mathcal{R}_j) \tag{25}$$

$$= \frac{1}{2n} \sum_i \mathcal{R}_i^2 + \frac{1}{2n} \sum_j \mathcal{R}_j^2 - \frac{1}{n^2} \sum_i \sum_j \mathcal{R}_i\mathcal{R}_j \tag{26}$$

$$= \frac{1}{n} \sum_i \mathcal{R}_i^2 - \left(\frac{1}{n} \sum_i \mathcal{R}_i\right)^2 \tag{27}$$

$$= \text{Var}(\mathcal{R}). \tag{28}$$

## G  Further results and details for experiments mentioned in main text

### G.1  CMNIST with covariate shift

Here we present the following additional results:

1. Figure 1 of the main text with additional results using MM-REx, see G.1. These results used the "default" parameters from the code of Arjovsky et al. (2019).

2. A plot with results on these same tasks after performing a random search over hyperparameter values similar to that performed by Arjovsky et al. (2019).

3. A plot with the percentage of the randomly sampled hyperparameter combinations that have satisfactory ($> 50\%$) accuracy, which we count as "success" since this is better than random chance performance.

These results show that REx is able to handle greater covariate shift than IRM, given appropriate hyperparameters. Furthermore, when appropriately tuned, REx can outperform IRM in situations with covariate shift. The lower success rate of REx for high values of $p$ is because it produces degenerate results (where training accuracy is less than test accuracy) more often.

The hyperparameter search consisted of a uniformly random search of 340 samples over the following intervals of the hyperparameters:

1. HiddenDim = [2**7, 2**12]

2. L2RegularizerWeight = [10**-2, 10**-4]

3. Lr = [10**-2.8, 10**-4.3]

4. PenaltyAnnealIters = [50, 250]

5. PenaltyWeight = [10**2, 10**6]

6. Steps = [201, 601]

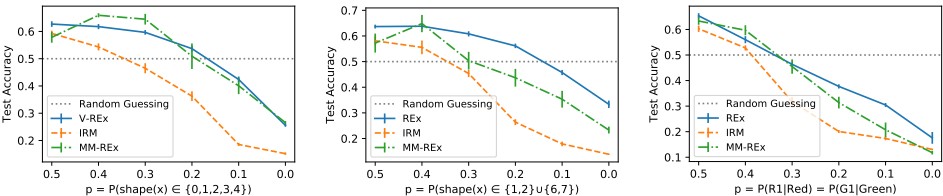

Figure 10: This is Figure 4 of main text with additional results using MM-REx. For each covariate shift variant (class imbalance, digit imbalance, and color imbalance from left to right as described in "CMNIST with covariate shift" subsubsection of Section 4.1 in main text) of CMNIST, the standard error (the vertical bars in plots) is higher for MM-REx than for V-REx.

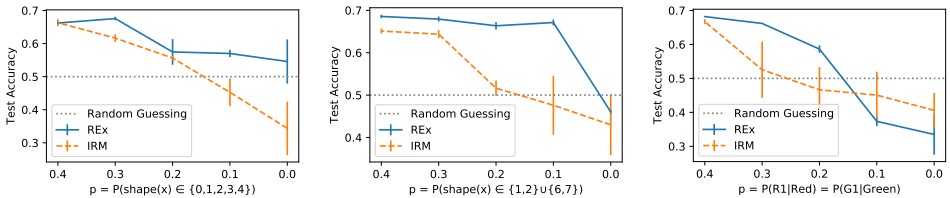

Figure 11: This is Figure 4 of main text (class imbalance, digit imbalance, and color imbalance from left to right as described in "CMNIST with covariate shift" subsubsection of Section 4.1 in main text), but with hyperparameters of REx and IRM each tuned to perform as well as possible for each value of p for each covariate shift type.

## G.2  SEMs from "Invariant Risk Minimization"

Here we present experiments on the (linear) structural equation model (SEM) tasks introduced by Arjovsky et al. (2019). Arjovsky et al. (2019) construct several varieties of SEM where the task is to predict targets $Y$ from inputs $X_1, X_2$, where $X_1$ are (non-anti-causal) causes of $Y$, and $X_2$ are (anti-causal) effects of $Y$. We refer the reader to Section 5.1 and Figure 3 of Arjovsky et al. (2019) for more details. We use the same experimental settings as Arjovsky et al. (2019) (except we only run 7 trials), and report results in Table 4.

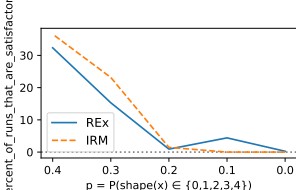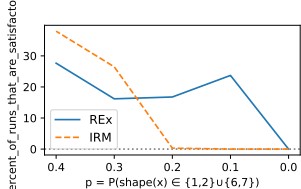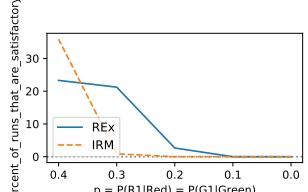

Figure 12: This also corresponds to class imbalance, digit imbalance, and color imbalance from left to right as described in "CMNIST with covariate shift" subsubsection of Section 4.1 in main text; but now the y-axis refers to what percentage of the randomly sampled hyperparameter combinations we deemed to to be satisfactory. We define satisfactory as simultaneously being better than random guessing and having train accuracy greater than test accuracy. For p less than .5, a larger percentage of hyperparameter combinations are often satisfactory for REx than for IRM; for p greater than .5, a larger percentage of hyperparameter combinations are often satisfactory for IRM than for REx because train accuracy is greater than test accuracy for more hyperparameter combinations for IRM. We stipulate that train accuracy must be greater than test accuracy because test accuracy being greater than train accuracy usually means the model has learned a degenerate prediction rule such as "not color".

These experiments include several variants of a simple SEM, given by:

$$X_1 = N_1$$
$$Y = W_{1 \to Y} X_1 + N_Y$$
$$X_2 = W_{Y \to 2} Y + N_2$$

Where $N_1, N_Y, N_2$ are all sampled i.i.d. from normal distributions. The variance of these distributions may vary across domains.

While REx achieves good performance in the **domain-homoskedastic** case, it performs poorly in the **domain-heteroskedastic** case, where the amount of intrinsic noise, $\sigma_y^2$ in the target changes across domains.[14] Intuitively, this is because the irreducible error varies across domains in these tasks, meaning that the risk will be larger on some domains than others, even if the model's predictions match the expectation $\mathbb{E}(Y|Pa(Y))$. We tried using a "baseline" (see Eqn. 4) of $r_e = Var(Y_e)$ (Meinshausen et al., 2015) to account for the different noise levels in $Y$, but this did not work.

We include a mathematical analysis of the simple SEM given above in order to better understand why REx succeeds in the domain-homoskedastic, but not the domain-heteroskedastic case. Assuming that $Y, X_1, X_2$ are scalars, this SEM becomes

$$X_1 = N_1$$
$$Y = w_{1 \to y} N_1 + N_Y$$
$$X_2 = w_{y \to 2} w_{1 \to y} N_1 + w_{y \to 2} N_Y + N_2$$

We consider learning a model $\hat{Y} = \alpha X_1 + \beta X_2$. Then the residual is:

$$\hat{Y} - Y = (\alpha + w_{1 \to y}(\beta w_{y \to 2} - 1)) N_1 + (\beta w_{y \to 2} - 1) N_Y + \beta N_2$$

Since all random variables have zero mean, the MSE loss is the variance of the residual. Using the fact that the noise $N_1, N_Y, N_2$ are independent, this equals:

$$\mathbb{E}[(\hat{Y} - Y)^2] = (\alpha + w_{1 \to y}(\beta w_{y \to 2} - 1))^2 \sigma_1^2 + (\beta w_{y \to 2} - 1)^2 \sigma_Y^2 + \beta^2 \sigma_2^2$$

Thus when (only) $\sigma_2$ changes, the only way to keep the loss unchanged is to set the coefficient in front of $\sigma_2$ to 0, meaning $\beta = 0$. By minimizing the loss, we then recover $\alpha = w_{1 \to y}$; i.e. in the domain-homoskedastic setting, the loss equality constraint of REx yields the causal model. On the other hand, if (only) $\sigma_Y$ changes, then REx enforces $\beta = 1/w_{y \to 2}$, which then induces $\alpha = 0$, recovering the *anti*causal model.

---

[14]See Footnote 12.

|  | FOU(c) | FOU(nc) | FOS(c) | FOS(nc) |
|---|---|---|---|---|
| IRM | 0.001±0.000 | 0.001±0.000 | 0.001±0.000 | 0.000±0.000 |
| REx, $r_e = 0$ | 0.001±0.000 | 0.008±0.002 | 0.007±0.002 | 0.000±0.000 |
| REx, $r_e = \mathbb{V}(Y_e)$ | 0.816±0.149 | 1.417±0.442 | 0.919±0.091 | 0.000±0.000 |

|  | POU(c) | POU(nc) | POS(c) | POS(nc) |
|---|---|---|---|---|
| IRM | 0.004±0.001 | 0.006±0.003 | 0.002±0.000 | 0.000±0.000 |
| REx, $r_e = 0$ | 0.004±0.001 | 0.004±0.001 | 0.002±0.000 | 0.000±0.000 |
| REx, $r_e = \mathbb{V}(Y_e)$ | 0.915±0.055 | 1.113±0.085 | 0.937±0.090 | 0.000±0.000 |

|  | FEU(c) | FEU(nc) | FES(c) | FES(nc) |
|---|---|---|---|---|
| IRM | 0.0053±0.0015 | 0.1025±0.0173 | 0.0393±0.0054 | 0.0000±0.0000 |
| REx, $r_e = 0$ | 0.0390±0.0089 | 19.1518±3.3012 | 7.7646±1.1865 | 0.0000±0.0000 |
| REx, $r_e = \mathbb{V}(Y_e)$ | 0.7713±0.1402 | 1.0358±0.1214 | 0.8603±0.0233 | 0.0000±0.0000 |

|  | PEU(c) | PEU(nc) | PES(c) | PES(nc) |
|---|---|---|---|---|
| IRM | 0.0102±0.0029 | 0.0991±0.0216 | 0.0510±0.0049 | 0.0000±0.0000 |
| REx, $r_e = 0$ | 0.0784±0.0211 | 46.7235±11.7409 | 8.3640±2.6108 | 0.0000±0.0000 |
| REx, $r_e = \mathbb{V}(Y_e)$ | 1.0597±0.0829 | 0.9946±0.0487 | 1.0252±0.0819 | 0.0000±0.0000 |

Table 4: Average mean-squared error between true and estimated weights on causal ($X_1$) and non-causal ($X_2$) variables. **Top 2:** When the level of noise in the anti-causal features varies across domains, REx performs well (FOU, FOS, POU, POS). **Bottom 2:** When the level of noise in the targets varies instead, REx performs poorly (FEU, FES, PEU, PES). Using the baselines $r_e = \mathbb{V}(Y)$ does not solve the problem, and indeed, hurts performance on the homoskedastic domains.

While REx (like ICP (Peters et al., 2016)) assumes the mechanism for $Y$ is fixed across domains (meaning $P(Y|Pa(Y))$ is independent of the domain, $e$), IRM makes the somewhat weaker assumption that $\mathbb{E}(Y|Pa(Y))$ is independent of domain. While it is plausible that an appropriately designed variant of REx could work under this weaker assumption, we believe forbidding interventions on $Y$ is not overly restrictive, and such an extension for future work.

## G.3 Reinforcement Learning Experiments

Here we provide details and further results on the experiments in Section 4.4. We take tasks from the Deepmind Control Suite (Tassa et al., 2018) and modify the original state, **s**, to produce observation, $\mathbf{o} = (\mathbf{s} + \epsilon, \eta \mathbf{s}')$ including noise $\epsilon$ and spurious features $\eta \mathbf{s}'$, where $\mathbf{s}'$ contains 1 or 2 dimensions of **s**. The scaling factor takes values $\eta = 1/2/3$ for the two training and test domains, respectively. The agent takes **o** as input and learns a representation using Soft Actor-Critic (Haarnoja et al., 2018) and an auxiliary reward predictor, which is trained to predict the next 3 rewards conditioned on the next 3 actions. Since the spurious features are copied from the state before the noise is added, they are more informative for the reward prediction task, but they do not have an invariant relationship with the reward because of the domain-dependent $\eta$.

The hyperparameters used for training Soft Actor-Critic can be found in Table 5. We used `cartpole_swingup` as a development task to tune the hyperparameters of penalty weight (chosen from $[0.01, 0.1, 1, 10]$) and number of iterations before the penalty is turned up (chosen from $[5000, 10000, 20000]$), both for REx and IRM. The plots with the hyperparameter sweep are in Figure 13.

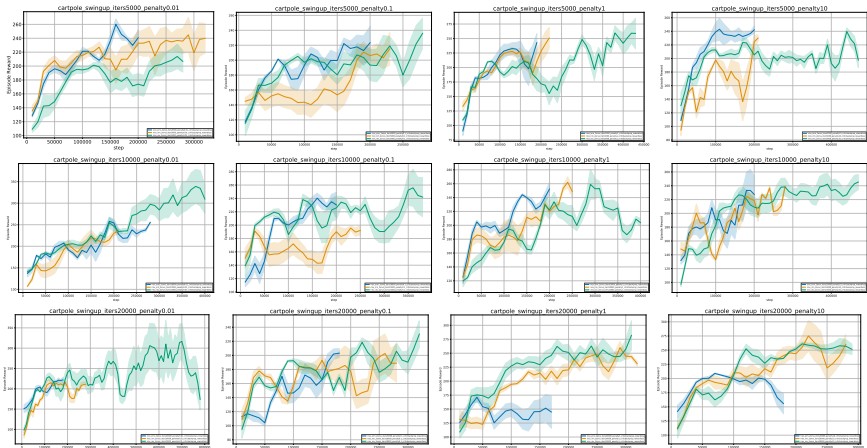

Figure 13: Hyperparameter sweep for IRM and REx on `cartpole_swingup`. Green, blue, and orange curves correspond to REx, ERM, and IRM, respectively. The subfigure titles state the penalty strength ("penalty") and after how many iterations the penalty strength was increased ("iters"). We chose a penalty factor of 1 and 10k iterations.

| Parameter name | Value |
|---|---|
| Replay buffer capacity | 1000000 |
| Batch size | 1024 |
| Discount $\gamma$ | 0.99 |
| Optimizer | Adam |
| Critic learning rate | $10^{-5}$ |
| Critic target update frequency | 2 |
| Critic Q-function soft-update rate $\tau_{\mathrm{Q}}$ | 0.005 |
| Critic encoder soft-update rate $\tau_{\mathrm{enc}}$ | 0.005 |
| Actor learning rate | $10^{-5}$ |
| Actor update frequency | 2 |
| Actor log stddev bounds | $[-5, 2]$ |
| Encoder learning rate | $10^{-5}$ |
| Decoder learning rate | $10^{-5}$ |
| Decoder weight decay | $10^{-7}$ |
| L1 regularization weight | $10^{-5}$ |
| Temperature learning rate | $10^{-4}$ |
| Temperature Adam's $\beta_1$ | 0.9 |
| Init temperature | 0.1 |

Table 5: A complete overview of hyperparameters used for reinforcement learning experiments.

## H  EXPERIMENTS NOT MENTIONED IN MAIN TEXT

We include several other experiments which do not contribute directly to the core message of our paper. Here is a summary of the take-aways from these experiments:

1. Our experiments in the CMNIST domain suggest that the IRM/V-REx penalty terms should be amplified exactly when the model starts overfitting training distributions.

2. Our financial indicators experiments suggest that IRM and REx often perform remarkably similarly in practice.

### H.1  A POSSIBLE APPROACH TO SCHEDULING IRM/REx PENALTIES

We've found that REx and IRM are quite sensitive to the choice of hyperparameters. In particular, hyperparameters controlling the scheduling of the IRM/V-REx penalty terms are of critical importance.

For the best performance, the penalty should be increased the relative weight of the penalty term after approximately 100 epochs of training (using a so-called "waterfall" schedule (Desjardins et al., 2015)). See Figure 14(b) for a comparison. We also tried an exponential decay schedule instead of the waterfall and found the results (not reported) were significantly worse, although still above 50% accuracy.

Given the methodological constraints of out-of-distribution generalization mentioned in (Gulrajani & Lopez-Paz, 2020), this could be a significant practical issue for applying these algorithms. We aim to address this limitation by providing a guideline for when to increase the penalty weight, based only on the training domains. We hypothesize that successful learning of causal features using REx or IRM should proceed in two stages:

1. In the first stage, predictive features are learned.

2. In the second stage, causal features are selected and/or predictive features are fine-tuned for stability.

This viewpoint suggests that we could use overfitting on the *training* tasks as an indicator for when to apply (or increase) the IRM or REx penalty.

The experiments presented in this section provide *observational* evidence consistent with this hypothesis. However, since the hypothesis was developed by observing patterns in the CMNIST training runs, it requires further experimental validation on a different task, which we leave for future work.

### H.1.1 RESULTS AND INTERPRETATION

In Figure 14, we demonstrate that the optimal point to apply the waterfall in the CMNIST task is after predictive features have been learned, but before the model starts to memorize training examples. Before predictive features are available, the penalty terms push the model to learn a constant predictor, impeding further learning. And after the model starts to memorize, it become difficult to distinguish anti-causal and causal features. This second effect is because neural networks often have the capacity to memorize all training examples given sufficient training time, achieving and near-0 loss (Zhang et al., 2016). In the limits of this memorization regime, the differences between losses become small, and gradients of the loss typically do as well, and so the REx and IRMv1 penalties no longer provide a strong or meaningful training signal, see Figure 15.

### H.2 DOMAIN GENERALIZATION: VLCS AND PACS

Here we provide earlier experiments on the VLCS and PACS dataset. We removed these experiments from the main text of our paper in favor of the more complete DomainBed results.

To test whether REx provides a benefit on more realistic domain generalization tasks, we compared REx, IRM and ERM performance on the VLCS (Torralba & Efros, 2011) and PACS (Li et al., 2017) image datasets. Both datasets are commonly-used for multi-source domain generalization. The task is to train on three domains and generalize to a fourth one at test time.

Since every domain in PACS is used as a test set when training on the other three domains, it is not possible to perform a methodologically sound evaluation on PACS after examining results on *any* of the data. Thus to avoid performing any tuning on test distributions, we use VLCS to tune hyperparameters and then apply these exact same settings to PACS and report the final average over 10 runs on each domain.

We use the same architecture, training procedure and data augmentation strategy as the (formerly) state-of-the-art Jigsaw Puzzle approach (Carlucci et al., 2019) (except with IRM or V-REx intead of JigSaw as auxilliary loss) for all three methods. As runs are very noisy, we ran each experiment 10 times, and report average test accuracies extracted at the time of the highest validation accuracy on each run. Results on PACS are in Table 7 while detailed results and performance on VLCS are left to the Appendix. On PACS we found that REx outperforms IRM and IRM outperforms ERM on average, while all are worse than the state-of-the-art Jigsaw method.

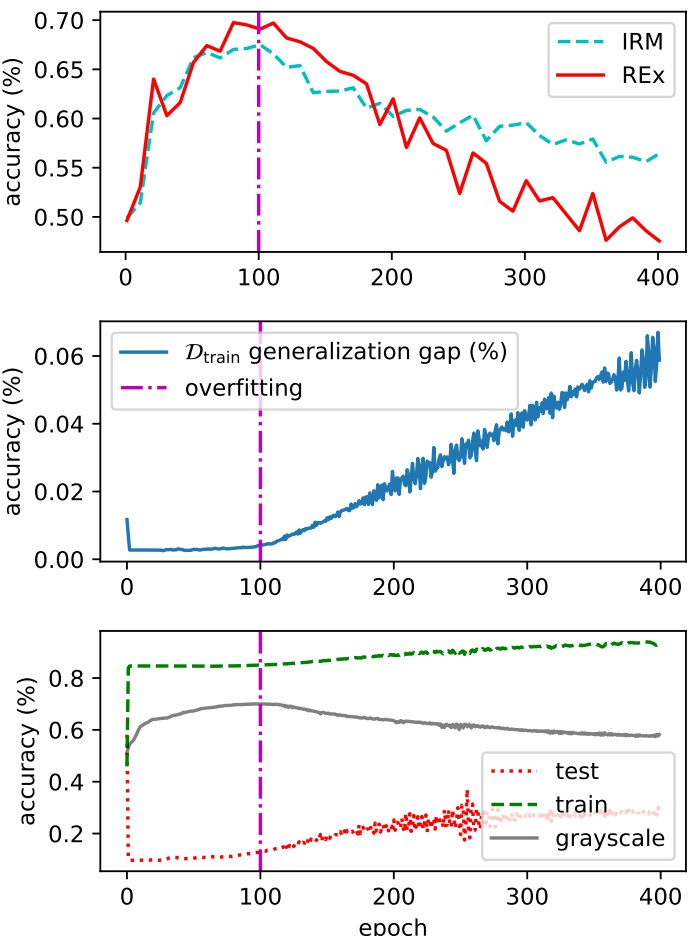

Figure 14: Stability penalties should be applied around when traditional overfitting begins, to ensure that the model has learned predictive features, and that penalties still give meanful training signals. **Top:** Test accuracy as a function of epoch at which penalty term weight is increased (learning rate is simultaneously decreased proportionally). Choosing this hyperparameter correctly is essential for good performance. **Middle:** Generalization gap on a validation set with 85% correlation between color and label (the same as the average training correlation). The best test accuracy is achieved by increasing the penalty when the generalization gap begins to increase. The increase clearly indicates memorization because color and shape are only 85%/75% correlated with the label, and so cannot be used to make predictions with higher than 85% accuracy. **Bottom:** Accuracy on training/test sets, as well as an auxilliary grayscale set. Training/test performance reach 85%/15% after a few epochs of training, but grayscale performance improves, showing that meaningful features are still being learned.

We use all hyperparameters from the original Jigsaw codebase.[15] We use Imagenet pre-trained AlexNet features and chose batch-size, learning rate, as well as penalty weights based on performance on the VLCS dataset where test performance on the holdout domain was used for the set of parameters producing the highest validation accuracy. The best performing parameters on VLCS were then applied to the PACS dataset without further changes. We searched over batch-sizes in $\{128, 384\}$, over penalty strengths in $\{0.0001, 0.001, 0.01, 0.1, 1, 10\}$, learning rates in $\{0.001, 0.01\}$ and used average performance over all 4 VLCS domains to pick the best performing hyperparameters. Table 6 shows results on VLCS with the best performing hyperparameters.

---

[15] https://github.com/fmcarlucci/JigenDG

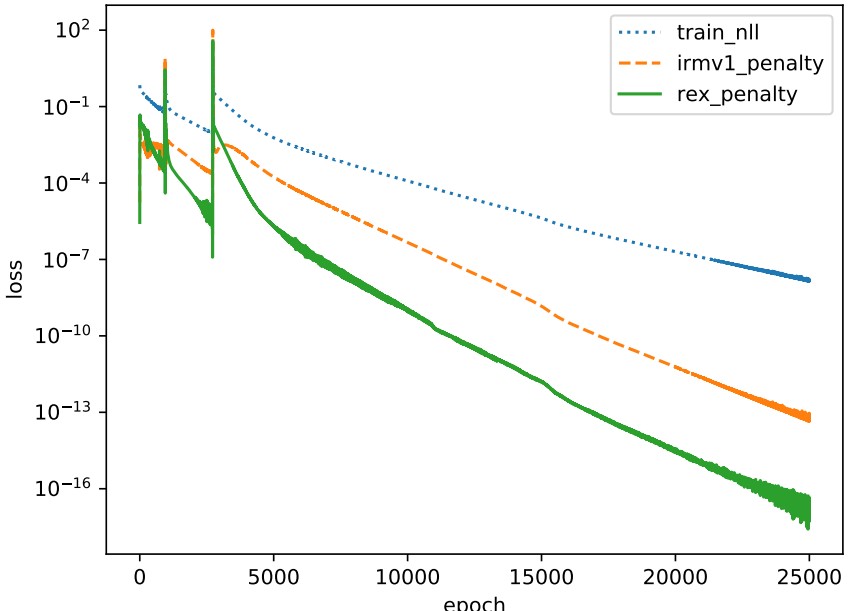

Figure 15: Given sufficient training time, empirical risk minimization (ERM) minimizes both REx and IRMv1 penalty terms on Colored MNIST (*without* including either term in the loss function). This is because the model (a deep network) has sufficient capacity to fit the training sets almost perfectly. This prevents these penalties from having the intended effect, once the model has started to overfit. The y-axis is in log-scale.

The final parameters for all methods on PACS were a batch size of 384 with 30 epochs of training with Adam, using a learning rate of 0.001, and multiplying it by 0.1 after 24 epochs (this step schedule was taken from the Jigsaw repo).The penalty weight chosen for Jigsaw was 0.9; for IRM and REx it was 0.1.We used the same data-augmentation pipeline as the original Jigsaw code for ERM, IRM, Jigsaw and REx to allow for a fair comparison.

| VLCS | CALTECH | SUN | PASCAL | LABELME | Average |
|---|---|---|---|---|---|
| **REx (ours)** | ~~**96.72**~~ | ~~63.68~~ | ~~**72.41**~~ | ~~60.40~~ | ~~**73.30**~~ |
| IRM | ~~95.99~~ | ~~62.85~~ | ~~71.71~~ | ~~59.61~~ | ~~72.54~~ |
| ERM | ~~94.76~~ | ~~61.92~~ | ~~69.03~~ | ~~**60.55**~~ | ~~71.56~~ |
| Jigsaw (SOTA) | ~~96.46~~ | ~~**63.84**~~ | ~~70.49~~ | ~~60.06~~ | ~~72.71~~ |

Table 6: Accuracy (percent) of different methods on the VLCS task. Results are test accuracy at the time of the highest validation accuracy, averaged over 10 runs. On VLCS REx outperforms all other methods. Numbers are shown in strike-through because we selected our hyperparameters based on highest test set performance; the goal of this experiment was to find suitable hyperparameters for the PACS experiment.

| PACS | Art Painting | Cartoon | Sketch | Photo | Average |
|---|---|---|---|---|---|
| REx (ours) | $66.27\pm0.46$ | $68.8\pm0.28$ | $59.57\pm0.78$ | $89.60\pm0.12$ | 71.07 |
| IRM | $66.46\pm0.31$ | $68.60\pm0.40$ | $58.66\pm0.73$ | $89.94\pm0.13$ | 70.91 |
| ERM | $66.01\pm0.22$ | $68.62\pm0.36$ | $58.38\pm0.60$ | $89.40\pm0.18$ | 70.60 |
| Jigsaw (SOTA) | $66.96\pm0.39$ | $66.67\pm0.41$ | $61.27\pm0.73$ | $89.54\pm0.19$ | 71.11 |

Table 7: Accuracy (percent) of different methods on the PACS task. Results are test accuracy at the time of the highest validation accuracy, averaged over 10 runs. REx outperforms ERM on average, and performs similar to IRM and Jigsaw (the state-of-the-art).

## H.3 FINANCIAL INDICATORS

We find that IRM and REx seem to perform similarly across different splits of the data in a prediction task using financial data. The dataset is split into five years, 2014–18, containing 37 publicly reported financial indicators of several thousand publicly listed companies each. The task is to predict if a company's value will increase or decrease in the following year (see Appendix for dataset details.) We consider each year a different domain, and create 20 different tasks by selecting all possible combinations of domains where three domains represent the training sets, one domain the validation set, and another one the test set. We train an MLP using the validation set to determine an early stopping point, with $\beta = 10^4$. The per-task results summarized in fig. 16 indicate substantial differences between ERM and IRM, and ERM and REx. The predictions produced by IRM and REx, however, only differ insignificantly, highlighting the similarity of IRM and REx. While performance on specific tasks differs significantly between ERM and IRM/REx, performance averaged over tasks is not significantly different.

### H.3.1 EXPERIMENT DETAILS

We use `v1` of the dataset published on [16] and prepare the data as described in.[17] We further remove all the variables that are not shared across all 5 years, leaving us with 37 features, and whiten the data through centering and normalizing by the standard deviation.

On each subtask, we train an MLP with two hidden layers of size 128 with tanh activations and dropout (p=0.5) after each layer. We optimize the binary cross-entropy loss using Adam (learning rate 0.001, $\beta_1 = 0.9$, $\beta_2 = 0.999$, $\epsilon = 10^{-8}$), and an L2 penalty (weight 0.001). In the IRM/REx experiments, the respective penalty is added to the loss ($\beta = 1$) and the original loss is scaled by a

---

[16]https://www.kaggle.com/cnic92/200-financial-indicators-of-us-stocks-20142018
[17]https://www.kaggle.com/cnic92/explore-and-clean-financial-indicators-dataset

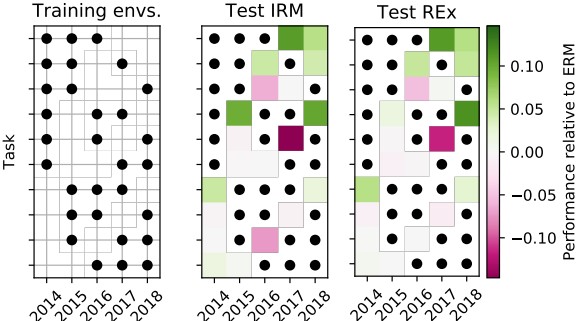

Figure 16: Financial indicators tasks. The left panel indicates the set of training domains; the middle and right panels show the test accuracy on the respective domains relative to ERM (a black dot corresponds to a training domain; a colored patch indicates the test accuracy on the respective domain.)

|  | Overall accuracy | Min acc. | Max acc. |
|---|---|---|---|
| ERM | $54.6 \pm 4.6$ | 47.6 | 66.2 |
| IRM | $55.3 \pm 5.9$ | 45.9 | 67.5 |
| REx | $\mathbf{55.5 \pm 6.0}$ | 47.2 | **68.0** |

Table 8: Test accuracy of models trained on the financial domain dataset, averaged over all 20 tasks, as well as min./max. accuracy across the tasks.

factor $10^{-4}$ after 1000 iterations. Experiments are run for a maximum of 9000 training iterations with early stopping based on the validation performance. All results are averaged over 3 trials. The overall performance of the different models, averaged over all tasks, is summarized in Tab. 8. The difference in average performance between ERM, IRM, and REx is not statistically significant, as the error bars are very large.

# I   OVERVIEW OF OTHER TOPICS RELATED TO OOD GENERALIZATION

**Domain adaptation** (Ben-David et al., 2010) shares the goal of generalizing to new distributions at test time, but allows some access to the test distribution. A common approach is to make different domains have a similar distribution of features (Pan et al., 2010). A popular deep learning method for doing so is Adversarial Domain Adaptation (ADA) (Ganin et al., 2016; Tzeng et al., 2017; Long et al., 2018; Li et al., 2018), which seeks a "invariant representation" of the inputs, i.e. one whose distribution is domain-independent. Recent works have identified fundamental shortcomings with this approach, however (Zhao et al., 2019; Johansson et al., 2019; Arjovsky et al., 2019; Wu et al., 2020).

Complementary to the goal of domain generalization is **out-of-distribution detection** (Hendrycks & Gimpel, 2016; Hendrycks et al., 2018), where the goal is to recognize examples as belonging to a new domain. Three common deep learning techniques that can improve OOD generalization are **adversarial training** (Goodfellow et al., 2014; Hendrycks & Dietterich, 2019), **self-supervised learning** (van den Oord et al., 2018; Hjelm et al., 2018; Hendrycks et al., 2019b; Albuquerque et al., 2020) and **data augmentation** (Krizhevsky et al., 2012; Zhang et al., 2017; Cubuk et al., 2018; Shorten & Khoshgoftaar, 2019; Hendrycks et al., 2019a; Carlucci et al., 2019). These methods can also been combined effectively in various ways (Tian et al., 2019; Bachman et al., 2019; Gowal et al., 2019). Data augmentation and self-supervised learning methods typically use prior knowledge such as 2D image structure. Several recent works also use prior knowledge to design augmentation strategies for invariance to superficial features that may be spuriously correlated with labels in object recognition tasks (He et al., 2019; Wang et al., 2019; Gowal et al., 2019; Ilse et al., 2020). In contrast, REx can discover which features have invariant relationships with the label without such prior knowledge.

