# OpenReview forum: "Out-of-Distribution Generalization via Risk Extrapolation (REx)"
_ICLR.cc/2021/Conference — Reject_

### Official Review · AnonReviewer1 · 2020-10-23
**A significant contribution to the field of domain adaptation and transfer learning.**

**Rating:** 6
**Confidence:** 2

**Review:**

Summary:
This paper addresses the problem of distributional shift in transfer learning from multiple training domains. The authors propose Risk Extrapolation (REx), which is a novel approach for out-of-distribution generalization when the new test domain for which we do not even have the covariate matrix. Thorough empirical experiments show that REx significantly outperforms state-of-the-art.

Pros:
- This is a highly quality paper with strong theoretical and empirical results.
- The paper is clearly written and easy to understand.
- Based on the thorough literature review, this idea of this work is original.
- The results of this work are highly significant and of interest to the domain adaptation and transfer learning community.

Cons:
- Although I understand the page limit, most of the major parts of the paper (especially the theoretical aspects) can be found in the appendix. As a person who is not very familiar with the literature on distributional shift from multiple domains, I did appreciate having this thorough overview; however, the detailed discussions of the contributions of the paper might be overlooked if (when) located in the appendix. Specifically, there is  only half a page on introducing the proposed methods for REx  in Sec. 3.1.

Minor comment(s):
- Reference “Peter Bühlmann. Invariance, causality and robustness, 2018a.” is duplicated.

############################################################

Post-Rebuttal:

After reviewing the concerns raised by the other reviewers, and the responses provided by the authors, I have decided to adjust my scores.

Moreover, I was disappointed that the authors did not use the extra one page to move some material from the appendix to the main text in order to elaborate on the proposed method.

---

> ### Author Response · Authors · 2020-11-12
> **On splitting content between main text and appendix**
>
> Thanks for your review.
>
> We found it hard to decide which material to leave out of the main text.
> Ultimately, we decided to focus on communicating the key ideas and intuitions of the paper with maximal clarity in the main text.
> We'd love any more specific suggestions for which content you'd like to see cut from or added to the main text.
> In particular, other reviewers suggested cutting Algorithm 1 and Figure 3, and including theoretical results from Appendix E.  What do you think of those suggestions?

---

### Official Review · AnonReviewer3 · 2020-10-26
**Official Blind Review #3**

**Rating:** 5
**Confidence:** 4

**Review:**

This paper studies the out-of-distribution (OOD) generalisation problem via risk extrapolation (REx). The authors propose two methods, MM-REx and V-REx, and empirically show that REx can recover the causal mechanisms on Colored MNIST, while also providing some robustness to covariate shift. The authors deal with the relationship between robustness, invariance and causality carefully, and provide experimental evidence beyond Colored MNIST.

I vote for weakly rejecting. This paper reviews various previous work but does not provide a clear comparison from them.

I have some comments and questions as follows:
1. Contribution 1 and Table 1 state that REx is suitable for invariant prediction. In the experiments, the authors also take IRM as the competitor. So I think the main objective of REx is the invariant prediction, rather than covariate shift. However, the authors emphasize that REx can deal with covariate shift. Please explain more on how REx discovers the invariant prediction.
2. When considering the invariant prediction, we aim to discover a stable conditional distribution $P(Y|X)$ or an invariant conditional mean $E[Y|X].$ However, the covariate shift refers to the changes in the distribution of X. What is the novelty of assuming covariate shift here? Please also figure out the importance of considering covariate shift under the invariant prediction.
3. What is the expression (1) in Page 1? If (1) is the risk function of the OOD generalization problem, $\mathcal{F}$ is unseen and should be the same throughout this paper. At the end of Page 1, you say: "Our method minimax Risk Extrapolation (MM-REx) is an extension of DRO where $\mathcal{F}$ instead contains affine combinations of training risks, see Figure 1." Does MM-REx solve a different OOD problem? Please figure out the definitions of $\mathcal{F}$ of  DRO, MM-REx and IRM respectively.
4. Please explain Contribution 3. In general, the equality of risks is not a sufficient condition of the causality.
5. In the VLCS and PACS experiments, the evaluation is incorrect. In Section 4.3, the task is to train on three domains and generalize to the fourth one at test time. However, this test accuracy is not worst-case performance.  According to (1), the problem is to generalize to all four domains at test time and to find out the worst domain. Then Table 3 should report the average of the worst-domain accuracy.

---

> ### Author Response · Authors · 2020-11-12
> **Please clarify argument for rejection.**
>
> It looks to us like this is the main reason given for rejecting our submission:
> "This paper reviews various previous work but does not provide a clear comparison from them."
> Can you please elaborate on this critique?
> And/or explain what, if not this, are the main reasons for recommending rejection?
>
>
> To clarify our position:
> - We consider the methods listed in Table 1 to be our primary points of comparison.
> - This table is meant to provide a summary of the qualitative advantages of REx over these alternatives, i.e. a clear comparison.
> - Arjovsky et al. 2019 already demonstrated IRM's advantages over the other methods listed.
> - Our work focuses on comparing REx with IRM, and especially the advantage REx has in handling covariate shift.  We believe our experiments (4.1,4.2,4.4), and mathematical analyses (C.2) provide a clear comparison.

---

> > ### Comment · AnonReviewer3 · 2020-11-13
> > **Official Blind Review #3**
> >
> > Thank you for the response. You have addressed some of my concerns. Overall, this paper is well written. My reasons for penalizing score is as follows：
> >
> > **1 [My major concern]: This paper reviews various previous work but does not provide a clear comparison from them.**
> >
> > Let us focus on three methods: group DRO, IRM and REx.  Throughout this paper, you consider IRM as the main competitor and show that REx is sensitivity to covariate shift as an important qualitative difference from IRM that can help improve results. So the foundation of this work is that REx can do invariant prediction.
> >
> > IRM explains the invariant prediction from the view of causality. However, REx is an extension of the group distributional robustness. In your response, you use " robustness to spurious features" to interpret the invariant prediction. This is imprecise. REx is robust to the conditional distribution $P(Y|X)$ rather than spurious features. Notice that, in Table 1, group DRO cannot do invariant prediction. So I think the first task of this work is to explain: How REx discovers invariant prediction by introducing negative weights.
> >
> > **2 About the evaluation procedure**
> >
> > Figure 2 gives a good example. In your response, you say: "The unseen domain is used as a proxy for how well the model would generalize in practice to unseen domains." We should note that, for the experiments on Colored MNIST, it is possible to achieve 0.9 test accuracy on the test domain with $P(Y=0|color=red)=0.9$ while the test accuracy on two training domains is just 0.1 and 0.2. This is not the OOD generalization in Eq.(1).
> >
> > The experiments should respond to the original OOD problem (1).  For VLCS and PACS,  all four domains mimic the unseen $\mathcal{F}$. Then three domains are observed: two for training and one for validation.  Then the OOD generalization is to generalize to all four domains.

---

> > > ### Author Response · Authors · 2020-11-17
> > > **RE (1)**
> > >
> > > Thanks for continuing the discussion!
> > >
> > > 1) We're sorry our explanations for how REx can perform invariant prediction were not clear enough.
> > > You are correct in observing that REx can provide robustness to arbitrary directions of change in $P(Y|X)$ (or more generally, $P(X,Y)$, see Figure 1), provided training domains illustrate such shifts.  Thus REx has the potential to provide robustness in a wider range of circumstances than methods which focus on invariant prediction, such as IRM.  The greater flexibility of REx is also a selling point of our work, which we highlight be demonstrating that REx can provide robustness to covariate shift.
> > >
> > > Regarding invariant prediction in particular, robustness to spurious features is a special case of robustness to change in $P(X,Y)$, and so REx can provide robustness to spurious features if the training domains indicate their spuriousness (note that IRM also requires evidence of spuriousness, in the same form of differences in the conditional $P(Y|X)$ across domains/environments).  This leads to invariant prediction when all spurious features can be identified as such.
> > >
> > > We also prove that REx can perform causal discover under suitable conditions in Appendix E; the homoskedasticity assumption of E.1.2 can also be replaced by a "no covariate shift" assumption (we'll mention this in the revision).

---

> > > > ### Comment · AnonReviewer3 · 2020-11-19
> > > > **Official Blind Review #3**
> > > >
> > > > Thank you for the response.
> > > >
> > > > **To:** "Regarding invariant prediction in particular, robustness to spurious features is a special case of robustness to change in $P(X,Y)$ , and so REx can provide robustness to spurious features if the training domains indicate their spuriousness (note that IRM also requires evidence of spuriousness, in the same form of differences in the conditional $P(Y|X)$  across domains/environments). This leads to invariant prediction when all spurious features can be identified as such.
> > > >
> > > > **RE:** As far as I understand, the invariant prediction of REx comes from the robustness to change in $P(Y|X).$ The group DRO is also robust to the distributional change.  Could you explain why group DRO cannot discover an invariant predictor?

---

> > > > > ### Author Response · Authors · 2020-11-19
> > > > > **Why DRO fails to learn an invariant predictor**
> > > > >
> > > > > >  Could you explain why group DRO cannot discover an invariant predictor?
> > > > >
> > > > > Certainly!
> > > > > A more precise statement would be: "DRO typically does not learn an invariant predictor" (even when REx does). First, let's establish that this is in fact the case, using Colored MNIST as an example.
> > > > >
> > > > > Consider Figure 2 (left), and to simplify things, let's restrict ourselves to considering the case where $\beta=0$ (no REx) or $\beta=10000$ (REx).  An invariant predictor in CMNIST must ignore color, and training with REx leads the model to ignore color, achieving ~70% accuracy.  However, color is a more predictive feature than shape in *both* CMNIST training domains, and thus the worst-case optimal predictor across training domains (which is what DRO seeks) would use color, and achieve ~80% accuracy.  Thus DRO will not learn an invariant predictor in CMNIST.
> > > > >
> > > > > Nonetheless, you are correct that DRO can induce robustness to $P(Y|X)$.  However, in the case of CMNIST, $P(Y=0 | \mathrm{color=red}) \in \{0.1, 0.2\}$ for the training distributions.  Since DRO only considers *convex* combinations of training distributions, it will only be worst-case optimal for distributions with $P(Y=0 | \mathrm{color=red}) \in [0.1, 0.2]$.  Unfortunately (for DRO), since the test distribution has $P(Y=0 | \mathrm{color=red}) = 0.9$, DRO will not generalize to the CMNIST test set.
> > > > >
> > > > > Referencing our abstract, like REx, DRO assumes "that variation across training domains is representative of the variation we might encounter at test time", but only REx assumes "that *shifts at test time may be more extreme in magnitude*".
> > > > >
> > > > > Please let us know if you'd like us to elaborate on any of these points.

---

> > > > > > ### Comment · AnonReviewer3 · 2020-11-21
> > > > > > **About invariant prediction**
> > > > > >
> > > > > > Thanks for the timely reply.
> > > > > >
> > > > > > **To:** "Referencing our abstract, like REx, DRO assumes "that variation across training domains is representative of the variation we might encounter at test time", but only REx assumes "that shifts at test time may be more extreme in magnitude"."
> > > > > >
> > > > > > **Re:**
> > > > > >
> > > > > > 1. Group DRO is robust to change in P(Y|X).
> > > > > >
> > > > > > 2. REx is more robust than group DRO because REx considers more extreme shifts in P(Y|X).
> > > > > >
> > > > > > 3. You experimentally show that REx can discover invariant predictor on Colored MNIST.
> > > > > >
> > > > > > 4. IRM aims to use causal features to derive invariant preditor. This implies that comparing to REx, IRM is more robust to change in P(Y|X).
> > > > > >
> > > > > > 5. REx is robust to changes in P(X) while IRM cannot.
> > > > > >
> > > > > > Am I right by saying that? What is the relationship between invariant prediction and causal discover?

---

> > > > > > > ### Author Response · Authors · 2020-11-21
> > > > > > > **Relationship between Invariant Prediction and Causal Discovery**
> > > > > > >
> > > > > > > We agree with your list, except on point 4.  It is unclear to whether REx or IRM is more robust to change in P(Y|X).  And we believe IRM should be viewed as a method for invariant prediction, which is only equivalent to discovering the causes of Y.  The only clear qualitative difference between REx and IRM that we are aware of, in terms of how they generalize OOD is the one we focus on: REx promotes robustness to covariate shift, but as a consequence can fail on heteroskedastic data.
> > > > > > >
> > > > > > > We define invariant prediction as in Koyama and Yamaguchi (2020).  Invariant prediction can *require* causal discovery (in the sense of identifying the causes of Y, rather than identifying the entire causal of the data, e.g. which elements of X cause each other); this has been proven in Peters et al. (2015) for the case where X contains the parents of Y, and the perturbation set is generated via certain interventions on X.  For more general cases, we believe the relationship is still not entirely clear to the research community (although this area has received significant attention this year, and so there may be some new works we are not familiar with).  However, it is the case that sometimes an invariant predictor can use non-causal information; see top of page 5 of Koyama and Yamaguchi (2020) for an example.
> > > > > > >
> > > > > > > We hope this helps clarify things, and are happy to continue discussing.

---

> > > ### Author Response · Authors · 2020-11-18
> > > **RE (2)**
> > >
> > > All of our experiments with CMNIST (and variants) already use this evaluation procedure; we will make sure this is clear in revision.
> > >
> > > We plan to include results using this form of evaluation for DomainBed experiments, as well.
> > > A fair comparison with other methods will require rerunning them, since the DomainBed paper/repo does *not* use this evaluation procedure.
> > > Due to limited resources, we plan to compare only ERM and REx.
> > > We hope to finish these experiments by the end of the discussion phase and include them in the revision, but may not have sufficient resources.

---

> ### Author Response · Authors · 2020-11-12
> **RE comments and questions**
>
> Thank you for these detailed comments and questions.
> They identify some important points which we hope to clarify and address here and in our revision.
>
> 1. In general, REx is meant to help provide robustness to *whatever* forms of distributional shift are observed in training domains.  In particular, this could include robustness to covariate shift and/or robustness to spurious features (i.e. invariant prediction).  IRM is the obvious point of comparison for invariant prediction; we emphasize REx's sensitivity to covariate shift as an important qualitative difference from IRM that can help improve results.
> We show experimentally that REx can solve invariant prediction tasks (CMNIST (4.1) and our Reinforcement Learning experiments (4.4)).  We also prove that REx discovers an invariant predictor in somewhat restricted settings, in Appendix E.
>
> 2. The main novelty is that previous methods for invariant prediction (e.g. IRM) do not provide this robustness to covariate shift.  Our experiments show that this leads IRM to underperform in settings that involve both covariate shift and spurious features.  Robustness to covariate shift is important in practice because it is very common and there is no reason we expect it *not* to co-occur with spurious features.
> Thus if a method for invariant prediction cannot perform well in the presence of covariate shift (which our experiments indicate is true of IRM), then that method may fail to perform well in realistic settings.
>
> 3.
> We apologize for the confusion, and will clarify this in revision.
> Equation 1 serves two roles.
> First, it is introduced in a general form as the OOD problem definition, where F is typically unknown.
> Second, it is used as an objective function, replacing the true, unknown F with some proxy.  In DRO, this proxy is the set of training domains.  In MM-REx, it is the set of extrapolated domains (see eqn5 and Appendix E.2).  In IRM, F is considered as domains corresponding to "valid interventions" (see bottom of page 10 and definition 7 of Arjovsky et al. 2019); this definition is similar to that of ICP (Peters et al. 2015) which considers the set of domains corresponding to interventions that preserve the causal mechanism of Y.
> In Appendix E, we show a connection between equalizing risks (as in REx) and robustness to interventions.
>
> 4. We agree this is not true in general.  Some conditions are given in Appendix E.  We believe our experiments also suggest that REx can be a good approach to discovering causal structure in practice.
>
> 5. We follow the evaluation procedure of Gulrajani & Lopez-Paz 2020, which we consider a methodologically sound and meaningful evaluation (the focus of their paper is on proper methodology for OOD generalization).  As mentioned above, we don't know the true F.  The unseen domain is used as a proxy for how well the model would generalize in practice to unseen domains.

---

### Official Review · AnonReviewer4 · 2020-10-28
**An interesting approach, some comments clarifying the motivation and conditions for success**

**Rating:** 6
**Confidence:** 4

**Review:**

Overview:

The authors propose Risk Extrapolation (Rex) which is an invariance-based approach to domain generalization. The main idea is to go from worst-casing over a convex combination of domains to an affine set of domains (MM-Rex) or to penalize the variance (V-Rex). Evaluations compare to IRM and ERM and show uniform improvements.

Positives:

The paper has a well written and motivated introduction, and there's substantial added expository material in the supplement. I would maybe tone down on the amount of bolded text.

The arguments are pretty well-thought-out, including some discussion of the differences between causal recovery, invariant prediction, and domain generalization.

The method itself seems simple (in a good way), though I have some questions below. I also wonder if the variance version of the objective can be tied to DRO based approaches via the fact that DRO on a chi-squared perturbation ball is equivalent to variance regularization of the risk.

The method seems to work well overall compared to IRM, and although it only matches ERM in the domain generalization benchmark, I think this is still a decent result given that most methods underperformed ERM.

Negatives:

This is possibly a comment that refers to a paper that's too recent, so not addressing this comment won't affect my rating of the paper, but it seems worthwhile from a scientific perspective to address how recent negative theory results (Rosenfeld, Ravikumar, Risteski 2020) about IRM (and REx) affect the intro framing. In particular, I'd like to see the claim in the intro about how REx can extrapolate can be reconciled with the claims in the other paper that IRM and REx succeed under the same conditions as ERM.

As a note: I also don't think Williamson and Menon suggest the use of variance of risks. They explicitly state that the R_{sd} risk aggregator is not a fairness risk measure. The risks fulfilling the axioms in that paper are coherent risk measures, and I believe they will all fall under the category of RI methods through duality arguments.

I can believe that minimizing the max risk is a reasonable thing to do, but it seems like an added leap of faith to want to actively increase the risk of the other domains since minimax risk would naturally equalize the different domains (if it's possible to achieve equal risk). It would be nice to see a clearer justification for this behavior.

Having looked at the supplement, there's substantial and good material there, and I would maybe suggest that the authors cut something like figure 3 and algorithm 1 to bring back some more intuition and motivation about REx, maybe some result from section E.

Having looked at the Thm 1 result, wouldn't 3 interventions on every variable also recover the true beta with ERM? This result also depends very heavily on fixed noise across environments. I do think this is a neat result though, and it might be useful to use this to give additional intuition about REx in the main text.

Does E.1.2 require pointwise homoskedasticity ? that seems wildly strong for a general result... In classification, I think this means that the map from x → y has to be deterministic and in general, for any log-probably loss, this says that the entropies have to be identical everywhere.

Minor:

Might be worth re-stating what the methods are in the experiment section (RI, for example, is defined pretty early on).

DRO is usually expanded to distributionally robust optimization, not domain robust optimization.

Figure 3 seems unnecessary.

Is Epsilon_j in the statement of Theorem 1 a typo? is there a typo in the subscript of beta_j = beta_{0,j} for all j?

---

> ### Author Response · Authors · 2020-11-17
> **Response**
>
> Thanks for the review, and the questions and pointers.
>
> Some responses, ordered as in your review:
> * We agree that the variance-based penalty can be related to the DRO version, and plan to include this in our revision.
> * Thanks for mentioning the Rosenfeld, Ravikumar, Risteski paper.  We haven't had time to look at it yet, but are interested in discussing it further.  Could you perhaps elaborate a bit, e.g. characterize these negative results and/or point to specific parts of the paper?
> * We'll take a closer look at Williamson and Menon and get back to you on this point.
> * In fact, minimax does *not* equalize risk across different domains.  Figure 3 (right) provides a demonstration: beta=0 gives better minimax (training) risk, but *worse* test risk. This is because paying more attention to color reduces risk on *both* training domains, but also *increases* the difference between training risks, and the risk on the *test* domain. This example should help explain out intuition for enforcing *exact* equality, as well as why DRO fails to do so, and thus cannot serve as a method for invariant prediction. Please let us know if any of this is still unclear.
> * We'll move some of the results such as statements of theorems to the main text, and would love more suggestions as to which material from the appendix would be most valuable to include in the main text.
> * No, ERM is sensitive to the number of examples from different domains; see the Remark on page 18.  In particular, we can consider the case where $1-\epsilon$ of the data comes from the observational distribution (with no interventions), and $\epsilon$ of the data comes from other domains.  Then the ERM objective is easily dominated by the loss on the observational distribution.
> * Yes, the assumptions for E.1.2 are indeed strong, although I'm not sure exactly what you mean by "pointwise" homoskedasticity.  The homoskedasticity assumption could be replaced with a "no covariate shift" assumption.  Note that IRM doesn't require either assumption, but also (unlike REx) fails to provide robustness to covariate shift in the limited data regime.
> It is possible in principle to control for the effects of covariate shift in the infinite data/capacity ("realizable") case using variant of REx, but we were advised in a previous review cycle to remove this content, which is still somewhat a work-in-progress.  Ultimately, we seek a method of invariant prediction that provides robustness to covariate shift in the limited data/capacity case, which does not require such strong assumptions; it seems like such a method would need to distinguish between inputs x that have high loss due to 1) inherent noise vs. 2) underfitting.
> * Thanks for the minor comments; we'll implement the suggestions in our revision (except we might keep Figure 3, space permitting).

---

> > ### Author Response · Authors · 2020-11-25
> > **Regarding Williamson and Menon**
> >
> > We agree with your assessment of this related work, and will update the paper accordingly.

---

### Official Review · AnonReviewer2 · 2020-10-29
**Not convinced by the main claim of the paper**

**Rating:** 4
**Confidence:** 5

**Review:**

This manuscript studies the problem of domain generalization and proposes a method, dubbed Rex, for this purpose. The main movitation of this work over the invariant risk minimization (IRM) [1] paradigm is that IRM is not robust to covariate shift, while the authors claim that Rex can deal with both covariate shift and concept shift together. Although the argument that IRM is not robust to covariate shift in the feature space is true, out-of-domain generalization is not the original goal of IRM either. My main concern is that it is not clear to me why Rex could deal with both covariate and concept shift together, as from the optimization formulation in Eq. (6), neither invariant predictor nor invariant representation is enforced. In particular, no theoretical analysis is given to justify this claim.

Perhaps what adds more confusion to me is that, what's the meaning of negative probability since the authors allow the combination weight \lambda of different domains to be negative? As the authors have already pointed out on page 5 (Probabilities vs Risks), the risk is a linear functional of the joint distribution over X and Y, hence using affine combination in the risk functions from different domains directly translate to allowing the use of negative coefficient for probabilities. This is quite strange, since the mixture distribution is only a convex combination, not affine combination.

Furthermore, I also found some of the discussions in the related work section misleading:
-   "The first method for invariant prediction ... is IRM". This is not correct. As the authors have already realized, ICP [2] has been proposed in 2015, and the definition of IRM is essentially the same as ICP.

-   The discussions about invariant representations on page 4 are not accurate. Only P_e(\phi) (but not P_e(\phi | Y)) is called invariant representations, and only this one can fail domain adaption if the marginal label distributions differ. The C-ADA does not try to find invariant P_e(\phi | Y). Instead, it tries to find invariant P_e(\phi x \hat{Y}), where \hat{Y} is the classifier output, and this is precisely the reason why C-ADA also fails under different label distributions. See more discussions in [4]. In fact, it has recently been shown in [4] that matching P_e(\phi | Y) provably works for domain adaptation, see Theorem 3.1 of [4].

-   "Also, unlike Rex, IRM seeks to match E[Y | \phi], not the full P(Y | \phi)". Again, this discussion is misleading. For the purpose of out of domain generalization, the learner does not need to match the full distributions P(Y | \phi). Only the conditional mean matters. See Theorem 4.1 of [5] as well as Theorem 3.3 of [6] in the context of fairness.

-   About the discussion on fairness of equalizing risk across groups. In fact a sufficient condition for this goal has been proved in Theorem 3.3 of [6]. Given the close relationship between these two problems, I feel it's necessary to cite and have a discussion of this result here as well.


More detailed comments:
-   The reference of David et al., 2010 should be Ben-David et al. 2010 on page 2

-   I think the naming of invariant prediction on page 2 is not very accurate. To be precise, as long as the same hypothesis (classifier) is used over different domains, this amounts to be an invariant prediction rule. Instead, what IRM enforces is the invariant OPTIMAL predictor, i.e., invariant conditional means.

-   Eq. (3) is not correct: the second term is a weighted combination of samples from different domains, where the larger the sample size from a domain the more weight it has in the combination. However, the right most term has a wrong weight for a domain. The correct one should be |D_e| / \sum_e |D_e|.

-   On page 4, the citation of Pan et al. 2010 is not accurate. Invariant representations for domain adaptation is first proposed by [3].



[1]     Invariant Risk Minimization
[2]     Causal inference using invariant prediction: identification and confidence intervals
[3]     Unsupervised Domain Adaptation by Backpropagation
[4]     Domain Adaptation with Conditional Distribution Matching and Generalized Label Shift
[5]     On Learning Invariant Representation for Domain Adaptation
[6]     Inherent Tradeoffs in Learning Fair Representations

---

> ### Author Response · Authors · 2020-11-17
> **RE: main claim**
>
> Thank you for your review.
>
> As I understand it, you were not convinced that REx can provide robustness by encouraging both: 1) robustness to covariate shift and 2) invariant prediction.
> This is indeed a central claim of our paper, which we believe is well supported experimentally and conceptually.
>
> Responding to your statements in more detail:
>
> > Although the argument that IRM is not robust to covariate shift in the feature space is true, out-of-domain generalization is not the original goal of IRM either.
>
> We're not sure we've understood this argument.  Can you clarify what you mean by this? Our understanding is that out-of-*distribution* (OOD) generalization is the goal of IRM.
> Would you agree?  Did you mean something different by out-of-*domain* generalization?
>
> As support the claim that IRM *is* focused on OOD generalization, we offer this quote from the abstract of Arjovsky et al. 2019:
> "we leverage tools from causation to develop the mathematics of spurious and invariant correlations, in order to alleviate the excessive reliance of machine learning systems on data biases, allowing them to generalize to new test distributions."
> We also refer to Arjovsky's thesis (see: https://cs.nyu.edu/dynamic/reports/?type=PhD), titled "Out of Distribution Generalization in Machine Learning", which is simply a somewhat expanded version of Arjovsky et al., 2019.
>
> In this thesis they also note that IRM may not help in the "realizable" case, where X is already an invariant predictor.  In this case, P(Y|X) is fixed by assumption, and so only covariate shift occurs.  Their discussion of this setting indicates their recognition of covariate shift as an important problem in OOD generalization.  Arjovsky's thesis recognizes this as an apparent limitation of IRM, while providing some arguments for optimism that invariant prediction could help even in this realizable case.
>
> >  My main concern is that it is not clear to me why Rex could deal with both covariate and concept shift together, as from the optimization formulation in Eq. (6), neither invariant predictor nor invariant representation is enforced. In particular, no theoretical analysis is given to justify this claim.
>
> * First, we want to note that our experiments provide a demonstration of REx dealing with both covariate and concept shift.
> * Our work also describes and illustrates conceptually why equalizing risks (i.e. "flattening the risk plane") could be expected to provide robustness to the kinds of shifts encountered at training time.  In the case of MM-REx this also follows directly from the definition and the linearity of the risk in the P(X,Y).
> * Appendix C.1/2 also provides simple examples illustrating how REx can encourage robustness to both kinds of shift.
> * In terms of theory, Appendix E.1 proves that REx can perform causal discovery.  This has a close correspondance with invariant prediction, as discussed in Arjovsky et al. (2019), although as far as we know, the details of the relationship between these two has not been fully elaborated.
>
> We recognize that many of these important results we refer to are in the Appendix, and plan to emphasize these results more clearly in the main text in our revision.  We also welcome suggestions regarding which content would be most helpful to move to the main text.

---

> ### Author Response · Authors · 2020-11-17
> **RE: negative probabilities**
>
> We're not sure we understand your criticism regarding negative probabilities; can you please elaborate in light of our comment below, if this doesn't clear it up?
>
> Quasiprobabilities are simply signed measures that integrate to 1, and allow negative probability to be assigned to some examples.  As we illustrate in Figure 3, negative weights $\lambda$ do not *necessarily* lead to negative probabilities (in which case we REx can be considered exactly as an instance of DRO).  We acknowledge that negative probabilities may be a limitation for REx, and discuss this in more detail in Appendix E.2
>
> Nonetheless, a central point of our work is that extrapolating beyond convex combinations provides new generalization powers (2nd line of page 2).  One reason this is significant is because Arjovsky et al. (2019) motivated IRM by showing inadequacies of the DRO approach in such setting. Our work shows that a simple generalization of DRO can work as well as IRM, or even better.  Given it's effectiveness, we consider the strangeness of this approach a strength: this is what sets us apart from the vast existing literature on robust optimization / DRO.

---

> ### Author Response · Authors · 2020-11-17
> **RE: "misleading discussion"**
>
> We'll respond to each each of your 4 bullet points in order.
> Briefly, we agree that the 2nd was misleading (and will fix it!), but not the others.
>
> * We do not claim that IRM is the first method for invariant prediction. Our claim, verbatim, is: "The first method for invariant prediction *to be compatible with modern deep learning problems and techniques* is Invariant Risk Minimization (IRM)".
> We've added the emphasis, since this was left our in your original review, and significantly changes the meaning of the statement.
> We consider the following to be substantial and significant differences between IRM and ICP (already discussed by Arjovsky et al. 2019): 1) ICP assumes a linear model.  2) ICP relies on statistical hypothesis testing 3) ICP assumes that elements of X are either causal or non-causal, whereas IRM allows them to contain a mix of causal and non-causal elements.  These are the factors which make IRM, but not ICP, compatible with modern deep learning.  The nonlinear extension of ICP (Heinze-Deml et al. 2018) drops the linearity assumption, but it otherwise similar to ICP.
> If you disagree with any of this, please explain?
> Or if you believe that ICP is "compatible with modern deep learning problems and techniques", even given these differences, can you please say why?
> * Thank you for pointing this out.  We will look into the works you mention and revise our submission accordingly.
> * Far from misleading, we consider this a clear, precise, and correct statement about the differences of these methods.
> The next sentence describes how this difference can be an advantage for IRM over REx, so we are not saying this to try and make ourselves look good!  However, we disagree that "only the conditional mean matters." For instance, in risk-sensitive situations, we might like to match the true output distribution, not just its mean.
> * This paper seems to be about fairness, which doesn't seem relevant, since our goal is not fairness.  Indeed, we mention the fairness literature in related work to note the similarity of the methods *and* the difference of the goals.  Regarding theorem 3.3: At a glance, this looks like this theorem is a slight restatement of Theorem 4.1 of Zhao et al. 2019, which we already cite.  Can you please elaborate on why you think it is necessary to cite and discuss this work and this result?

---

> ### Author Response · Authors · 2020-11-17
> **RE: more detailed comments**
>
> * Thanks for noticing; we'll fix it!
> * We're not sure we understand this point.  To clarify, our use of the term "invariant predictor" is based on Koyama & Yamaguchi (2020).  It refers to a representation $\phi$ for which the *true* $P(Y|\phi)$ is invariant across environments.
> We would welcome better terminology, since $\phi$ is in fact a representation, not a predictor, but the term "invariant representation" is already taken.  In any case, this notion seems different from what you are describing, and is meant to define this concept, not to characterize what IRM does.
> * Thanks for noticing; we'll fix it!
> * We believe it is accurate, but are open to being corrected!  Pan et al. (2010) certainly predates Ganin et al. (2015) [3], and we believe Pan et al. introduce this idea.  Note that Ganin et al. themselves claim that Pan et al. "seek an explicit feature space transformation that would map source distribution into the target ones".

---

### Decision · Program_Chairs · 2021-01-07
**Final Decision**

**Decision:**

Reject

**Comment:**

The paper is proposing Risk Extrapolation (REX) as a domain generalization algorithm. Authors extends the distributionally robust learning to affine mixture of distributions from convex mixture. Authors later uses variances instead of this extension and demonstrate various empirical and theoretical properties. The paper is reviewed by four expert reviewers and the reviewers did not reach to a consensus. Hence, I also read the paper in detailed and reviewed it. In summary, reviewers argue the following:

- R#2: Main argument is the lack of justification of the claim "Rex could deal with both covariate and concept shift together". Authors try to address this in their response. Moreover, reviewer also argues in the private discussion that manuscript is not updated and authors did not address any of the issues during the discussion period.
- R#3: Argues that (similar to R#2), dealing with covariate shift is not explained properly. Reviewer is not persuaded that REX results in invariant prediction.
- R#1 and R#4: Largely positive about the paper. In the mean time, argue that organization of the paper is lacking and some of the material in the supplement is relevant and should be moved to the main text. R#1 decreases their score due to the lack of re-organization during the discussion.

The value of the paper is clear to me, the joint treatment of minimax perspective, domain generalization and invariances is definitely interesting and valuable. Hence, the paper has merit to be published. However, the presentation is lacking significantly.  The main contribution of the paper lies in Table 1 but the invariant prediction property is not justified at all in the main text. Hence, Table 1 is not justified properly. Authors discuss Thm 1&2 in their response but they both are in the supplement. From reading only the main text, confusion of the reviewers are well justified. ICLR guidelines clearly states that "...Note that reviewers are encouraged, but not required to review supplementary material during the review process..." It is authors' responsibility to make the main paper self contained. Even more worrisome is the fact that authors dismiss this concern in their response to R#1 which eventually leads to R#1 decreasing their score. Hence, I decided to reject the paper since the presentation is subpar and authors did not persuaded reviewers that they can fix this presentation issue by the camera-ready deadline. On the other hand, I think the paper can be really influential if it was written clearly. I suggest authors to revise the claims more precisely, extended the discussion on the claims and move the theorems to the main paper.